# HairDiffusion: Vivid Multi-Colored Hair Editing via Latent Diffusion

**Yu Zeng**[1], **Yang Zhang**[1*], **Jiachen Liu**[1], **Linlin Shen**[1,2,3]
**Kaijun Deng**[1], **Weizhao He**[1], **Jinbao Wang**[3,4]

[1]Computer Vision Institute, School of Computer Science & Software Engineering, Shenzhen University
[2]Shenzhen Institute of Artificial Intelligence and Robotics for Society
[3]National Engineering Laboratory for Big Data System Computing Technology, Shenzhen University
[4]Guangdong Provincial Key Laboratory of Intelligent Information Processing

{cengyu,liujiachen,dengkaijun}2023@email.szu.cn, heweizhao2022@email.szu.edu.cn
{yangzhang,llshen,wangjb}@szu.edu.cn

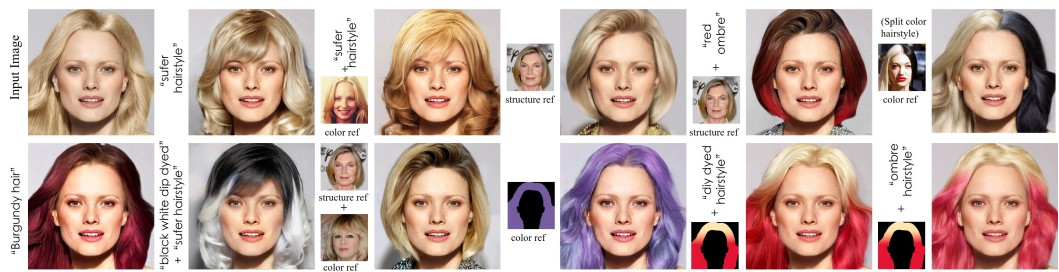

Figure 1: Our framework supports individual or collaborative editing of hairstyle and color, utilizing text, reference images, and stroke maps. With exceptional performance, particularly evident in editing multiple hair colors.

## Abstract

Hair editing is a critical image synthesis task that aims to edit hair color and hairstyle using text descriptions or reference images, while preserving irrelevant attributes (e.g., identity, background, cloth). Many existing methods are based on StyleGAN to address this task. However, due to the limited spatial distribution of StyleGAN, it struggles with multiple hair color editing and facial preservation. Considering the advancements in diffusion models, we utilize Latent Diffusion Models (LDMs) for hairstyle editing. Our approach introduces Multi-stage Hairstyle Blend (MHB), effectively separating control of hair color and hairstyle in diffusion latent space. Additionally, we train a warping module to align the hair color with the target region. To further enhance multi-color hairstyle editing, we fine-tuned a CLIP model using a multi-color hairstyle dataset. Our method not only tackles the complexity of multi-color hairstyles but also addresses the challenge of preserving original colors during diffusion editing. Extensive experiments showcase the superiority of our method in editing multi-color hairstyles while preserving facial attributes given textual descriptions and reference images.

## 1 Introduction

Hair editing is one of the most challenging and interesting facets of semantic face editing. The essence of this task is to edit hair attributes such as color, and hairstyle from the reference image or

---

*Corresponding author.

38th Conference on Neural Information Processing Systems (NeurIPS 2024).

text to the input image while preserving irrelevant attributes. Disentangling these attributes is the key to a quality solution to the problem. This task has many applications among professionals and amateurs during work with face editing programs, virtual reality, and computer games [20, 43, 37].

In recent years, numerous methods leveraging Generative Adversarial Networks (GANs) [33, 35, 14, 16, 36, 45, 17] have emerged in this field. These methods often involve mapping images into StyleGAN [16] latent space to disentangle hair features or utilizing Contrastive Language-Image Pre-Training (CLIP) model [27] to translate textual descriptions into relevant semantic vectors, facilitating manipulation via text or reference images.

However, previous methods have overlooked the **hair color structure**. As shown in Figure 1, "ombre hair" (the last column of the second row) means a hair color structure with a gradient transition from top-to-bottom, while "split color hair" (the last column of the first row) exhibits a hair color structure with a left-to-right transition. In traditional StyleGAN-based approaches [45, 35, 36, 17], the scarcity of multi-color hair in the facial datasets [15, 16] used to train StyleGAN [16], coupled with the limited distribution of latent space in StyleGAN [16], gives rise to two primary challenges: 1) difficulty in generating intricate hair color and hairstyle due to the insufficient diversity in the training data's multi-color hair distributions; 2) challenges in preserving the original facial information when editing the latent code after mapping images into latent space, leading to difficulties in editing images while preserving irrelevant attributes.

In recent years, with the advancement of diffusion models [39, 40, 47, 42, 3], their robust and stable generative capabilities have surpassed those of GANs in many aspects [6]. In particular, Latent Diffusion Models (LDMs) [12, 40, 47, 28] have demonstrated exceptional generative capabilities, notably in image inpainting tasks. However, the application of diffusion models to hair editing encounters three challenges: 1) lack of tailored masks for hairstyle inpainting, necessitating consideration of hairstyle regions while preserving irrelevant attributes; 2) difficulty in providing sufficient control for the hair editing task, which requires faithful transfer of hair color from another image or retaining the original hair color of the image; 3) limitations in text and semantic understanding related to hair color and hairstyle, hinder the precision of CLIP-guided diffusion processes.

To address these limitations, we propose a novel baseline approach based on LDMs, enabling the automatic generation of edited regions while allowing separate control of hair color and hairstyle via text and reference images individually or collaboratively. Our pipeline can be roughly divided into two stages. The first stage begins with obtaining hair-agnostic masks to acquire facial representations independent of hair. These representations are combined with facial keypoint information and processed through textual descriptions to perform the LDMs process, generating a style proxy (i.e., an image used to guide the hairstyle transfer process). The second stage involves transferring hair color using a reference image. Initially, the hair of the reference image is aligned with the source image using a pre-trained warping module. The aligned hair serves as a color proxy (i.e., an image used to guide the color and color structure transfer process), which is then input into the LDMs process to transfer or retain the hair color. Simultaneously, the Canny edge from the original hair or style proxy is employed to retain or edit the hairstyle. The key to the pipeline is to use different hair-agnostic masks for both the color proxy and style proxy at different stages and blend them in the diffusion latent space by Multi-stage Hairstyle Blend (MHB). By training the warping model to obtain aligned hair color prior information in targeted hair regions, our method effectively addresses multi-hair color editing tasks. Quantitative and qualitative evaluations, along with user studies, demonstrate the efficacy of our approach in hair editing tasks, particularly in color transformation and preservation of unrelated attributes within the diffusion model framework. Furthermore, by designing different stages of agnostic masks, we can maximize the retention of hairstyle-independent features such as background information and facial features. In summary, our contributions are outlined as follows:

- We present a warping module designed for hair warping, allowing the alignment of the target hair mask with precision and enabling comprehensive hair color structure editing through reference images.

- The MHB method is proposed within LDMs, which enables the decoupling of hair color and hairstyle, thereby effectively achieving high-quality hair color and hairstyle editing.

- Through extensive qualitative and quantitative evaluations, we showcase the superior performance of our method in text-based hairstyle editing, reference image-based hair color editing, and preservation of facial attributes.

- The application of LDMs to address the challenge of text and image-based hair editing is pioneered through the introduction of hair-agnostic facial representation masks, reframing hair editing as an inpainting task and representing a novel approach. To the best of our knowledge, this method has not been previously explored in this domain.

## 2  Related Work

**Diffusion Models.** Diffusion models have established themselves as a robust class of generative models capable of producing high-quality images through a process of iteratively denoising data [6, 12, 31, 28]. Specifically, SD-Inpainting [29] is based on the large-scale pre-trained text-to-image model, i.e., Stable Diffusion [28]. By utilizing random masks as repair masks and employing image-text prompts, diffusion models can fill areas with content consistent with textual descriptions while retaining context awareness in other regions. Despite the success of diffusion models in the inpainting task, their potential has not been fully exploited yet. For instance, describing fine details like hair color directly through text remains challenging. Some methods combine different interaction strategies to achieve satisfactory inpainting results, such as ControlNet-Inpainting [42] which utilizes ControlNet to guide image restoration based on additional inputs like Canny edge, depth maps, etc. Additionally, some methods provide an additional reference image or involve drawing rough color strokes. In the hair editing task, it is essential to develop a framework that facilitates multimodal conditions for image inpainting. Therefore, we leverage ControlNet to control hairstyle attributes and stroke maps, managing color information and color structure.

**Hair Editing.** As an essential component of the face, there have been numerous works on hairstyle editing and synthesis [33, 44, 45, 19, 35, 36]. Some works decouple facial attributes by using image-level masks and then splice hairstyle images onto facial images through generative network [33, 30]. However, this approach often results in inconsistencies in lighting or artificial shadows along the edges of the hairstyle region. Some approaches utilize the e4e [34] to map the target image and hairstyle reference image into $\mathcal{W}+$ latent space of StyleGAN [16], enabling manipulation of hairstyle and hair color via vector operations, followed by image synthesis using StyleGAN [35, 36, 26, 11]. HairCLIP [35] integrates CLIP [27] into hair-editing tasks by leveraging StyleGAN to decouple hair color and hairstyle features, enabling text-driven. Additionally, some methods attempt to enhance the generative capabilities of StyleGAN through various transformations of its latent space [1, 46, 45]. Despite these advancements, challenges persist due to the limited data distribution inherent in StyleGAN [16], which is trained predominantly on facial datasets [16, 23]. This constraint leads to difficulties in preserving specialized facial features like earrings and eyeshadow, as well as in introducing novel attributes such as multi-colored hairstyles. To tackle the constraints of hair color diversity, this paper introduces stroke maps [24] into the denoising process of diffusion models as the color proxy prior. Utilizing a CLIP text encoder and image encoder fine-tuned with images matching the hair color to achieve editing of hair color structure through text. For reference images, we finetune the model with textual inversion.

**Image Deformation.** Image deformation, or image warping, is a technique in computer vision and graphics for manipulating and transforming images [7, 32, 22]. The goal is to deform an input image while preserving its essential content and structure. In recent years, deep learning techniques have revolutionized the field of image deformation, particularly GANs [8], have revolutionized the field of image deformation [7, 32, 9]. GANs are employed to learn complex deformation mappings directly from data, capturing intricate image transformations and generating realistic deformations with minimal artifacts. Conditional GANs (cGANs) [25] represent an evolution in image deformation by incorporating additional contextual information during deformation. These networks learn to map input images to desired target outputs based on specific conditions or auxiliary information [32]. For example, integrating DensePose [10], detailing pixel-level semantic information enables precise and controlled image warping. In this paper, we employ multiple conditioning inputs to train a warping network capable of aligning hair color while accounting for the relationship between hair and facial features.

## 3  Method

As shown in Figure 2, we propose a novel solution based on LDMs for the first time in this field. In particular, our work employs the Stable Diffusion architecture as a starting point for performing

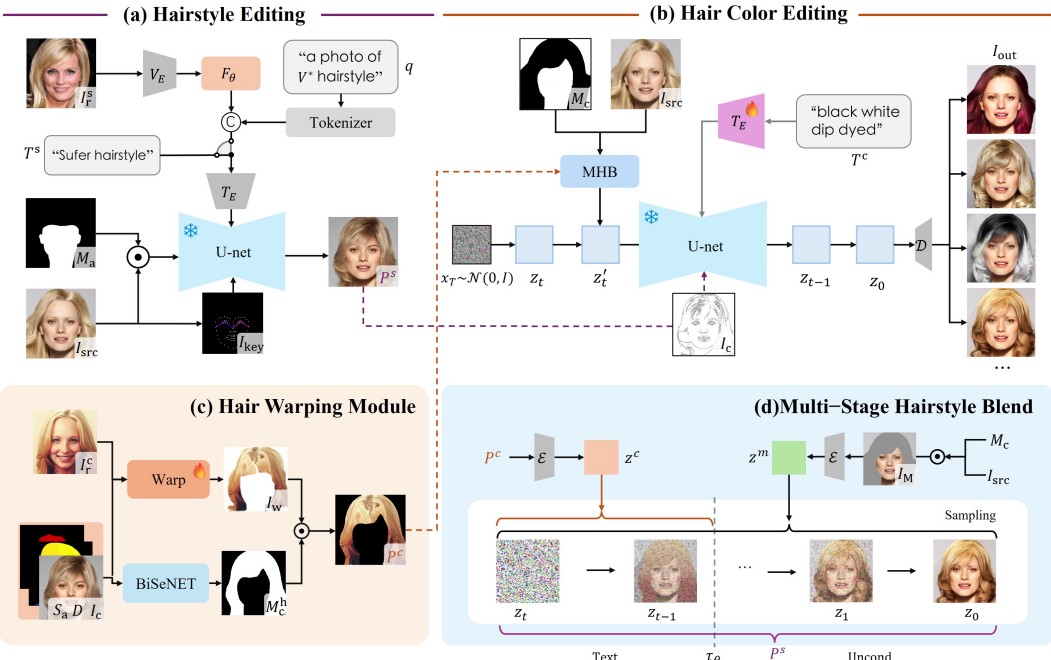

Figure 2: Overview of HairDiffusion: (a) Using a hairstyle description $T^{\text{s}}$ or reference image $I_{\text{r}}^{\text{s}}$ as conditional input, coupled with the hair-agnostic mask $M_{\text{a}}$ and source image $I_{\text{src}}$, we can get the style proxy $P^s$. (b) Leveraging the color proxy and style proxy, along with the hair-agnostic mask $M_{\text{c}}$ and source image $I_{\text{src}}$, enables individual or collaborative editing of hair color and hairstyle. (c) Given a series of conditions driven from the input image $I_{\text{c}}$, the hair color reference image $I_{\text{r}}^{\text{c}}$ is used to obtain the color proxy $P^c$ through a warping module. In the case of changing only the hairstyle while preserving the original hair color, $I_{\text{r}}^{\text{c}} = I_{\text{src}}$. (d) The color proxy $P^c$ and the style proxy $P^s$ are blended at different stages of the diffusion process.

the hairstyle editing task. We obtain the style proxy, denoted as $P^s$, to capture the hairstyle prior as shown in Figure 2 (a). Concurrently, we acquire the color proxy $P^c$, using a warping module as shown in Figure 2 (c). Finally, we incorporate these two proxies into the diffusion process through **Multi-stage Hairstyle Blend (MHB)** to blend them effectively as shown in Figure 2 (d). Therefore, our framework supports individual or collaborative editing of hairstyle and single-color or multi-color, utilizing text, and reference images.

## 3.1 Preliminaries

**Stable Diffusion.** It consists of an autoencoder encoder $\mathcal{E}$ and an autoencoder decoder $\mathcal{D}$, a text-conditional U-Net denoising model $\epsilon_\theta$, a CLIP text encoder $T_E$, which takes text $T$ as input. The encoder $\mathcal{E}$ compresses an image $I$ into the latent space of diffusion, while the decoder $\mathcal{D}$ performs the inverse operation and decodes a latent variable into the pixel space. For clarity, we refer to the $\epsilon_\theta$ convolutional input as the spatial input $\gamma$ since convolutions preserve the spatial structure, and to the attention conditioning input as $\psi$. The training of the denoising network $\epsilon_\theta$ is performed by minimizing the following loss function:

$$L = \mathbb{E}_{\mathcal{E}(I),\gamma,\epsilon\sim\mathcal{N}(0,1),t}\left[\left\|\epsilon - \epsilon_\theta(\gamma,\psi)\right\|_2^2\right], \tag{1}$$

where $t$ represents the diffusing time step, $\gamma = z_t$, $z_t$ is the encoded image $\mathcal{E}(I)$ where we stochastically add Gaussian noise $\epsilon \sim \mathcal{N}(0,1)$, and $\psi = [t; T_E(T)]$. Our goal is to generate a new image $I_{\text{out}}$, editing the hair based on user-provided reference images or textual descriptions of hairstyle or hair color, while preserving the unrelated features of the facial region outside the hair. This task can be viewed as a special kind of inpainting, specifically replacing the hair information in a face image based on user-provided conditions. Therefore, we use the Stable Diffusion inpainting pipeline as the starting point of our approach. It takes as spatial input $\gamma$ the channel-wise concatenation of an encoded masked image $\mathcal{E}(I_{\text{M}})$, a resized binary inpainting mask $m \in \{0,1\}^{1 \times h \times w}$, and the denoising network input $z_t$. $I_{\text{M}}$ is the input image $I_{\text{src}}$ masked according to the inpainting mask

$M_{\mathrm{c}} \in \{0, 1\}^{1 \times H \times W}$, and the binary inpainting mask $m$ is the resized version according to the latent space spatial dimension of the original inpainting mask $M_{\mathrm{c}}$. The spatial input of the inpainting denoising network is $\gamma = [z_t; m; \mathcal{E}(I_{\mathrm{M}})] \in \mathbb{R}^{(4+1+4) \times h \times w}$.

**ControlNet.** It is an extension of the diffusion model that incorporates conditional control to generate high-quality images with specific attributes. The main objective is to leverage additional conditional information $c$ (e.g., Canny edge, Openpose, Depth Map) to steer the image generation process. Combined with spatial input $\gamma$ (e.g., $z_t$), the diffusion process can be expressed as:

$$p_\theta(z_{t-1}|z_t, c) = \mathcal{N}(z_{t-1}; \mu_\theta(z_t, t, c), \Sigma_\theta(z_t, t, c)), \tag{2}$$

Both $\mu_\theta$ and $\Sigma_\theta$ depend on the conditional information $c$. We introduce the pose image $I_{\mathrm{key}}$, obtained from $I_{\mathrm{src}}$ using a pose keypoint extraction model [4], during the hairstyle editing stage to align the generated hair with the face in $P^s$. In the hair color editing stage, we use an edge detection model to obtain $I_{\mathrm{c}}$ from $P^s$ or $I_{\mathrm{src}}$ to guide the generation of the hairstyle.

**CLIP.** It is a vision-language model [27], which aligns visual and textual inputs in a shared embedding space. CLIP consists of an image encoder $V_E$ and a text encoder $T_E$ that extract feature representations $V_E(I) \in \mathbb{R}^d$ and $T_E(E_L(T)) \in \mathbb{R}^d$ for an input image $I$ and its corresponding text caption $T$, respectively. Here, $d$ is the size of the CLIP embedding space, and $E_L$ is the embedding lookup layer which maps each $T$ tokenized word to the token embedding space $\mathcal{W}$. To tackle the text-multi-color hair editing task, we utilize hair color structure text $T_{\mathrm{m}}$ in comparison with the multi-color hairstyle image $I_{\mathrm{m}}$, scraped from the internet to fine-tune the CLIP. The fine-tuning process involves aligning the texts with the corresponding images. To enable the model to learn the color structure of hair, we perform data augmentation through rotation and symmetry operations, introducing a variety of directional patterns. The fine-tuning objective is given by: $\min_\theta \mathbb{E}_{(T_{\mathrm{m}}, I_{\mathrm{m}}) \sim \mathcal{D}_{\mathrm{aug}}} [\mathcal{L}(T_{\mathrm{m}}, I_{\mathrm{m}}; \theta)]$ where the augmented dataset $\mathcal{D}_{\mathrm{aug}}$ is defined as: $\mathcal{D}_{\mathrm{aug}} = \{\mathrm{Augment}(\mathbf{x}_i) \mid \mathbf{x}_i \in \mathcal{D}\}$, $\theta$ represents the parameters of the CLIP text encoder $T_E$.

For reference image-based hairstyle editing, we employ textual inversion. Initially, we construct a textual prompt $q$ that guides the diffusion process. This prompt is tokenized and each token is mapped into the token embedding space using the CLIP embedding lookup module, resulting in $V^*$. Next, we encode the reference image $I_{\mathrm{r}}^{\mathrm{s}}$ using the CLIP visual encoder $V_E$, feeding the features extracted from the last hidden layer into the textual inversion adapter $F_\theta$. This adapter maps the input visual features to the CLIP token embedding space $\mathcal{W}$. The final prompt embedding vectors, combined with the predicted pseudo-word token embeddings, are formulated as follows:

$$E = \mathrm{Concat}(V^*, F_\theta(V_E(I_{\mathrm{r}}^{\mathrm{s}}))). \tag{3}$$

The concatenated embedding $E$ is then fed into the CLIP text encoder $T_E$, and the output is used to condition the denoising network by leveraging the existing Stable Diffusion textual cross-attention mechanism. We train $F_\theta$ with the input of a hair-agnostic masked face $\mathcal{E}(I_M)$, compared with face-agnostic mask $M_{\mathrm{a}}$, and the latent variable $z$. During the training of the adapter $F_\theta$, the parameters of both CLIP and the U-net are kept frozen.

## 3.2 HairDiffusion

**Data Preparation.** We define the hair-agnostic masks for inpainting in the two stages. In the hair editing stage, the $M_{\mathrm{a}}$ retains facial information (excluding the forehead area) and neck information while removing other irrelevant details(background, hair, etc). In the color editing stage, the mask $M_{\mathrm{c}}$ is used to remove the hair information, $M_{\mathrm{c}} = M_{\mathrm{h}} \cup M_{\mathrm{p}}$, $M_{\mathrm{h}}$ denotes $I_{\mathrm{src}}$ hair region mask, $M_{\mathrm{p}}$ denotes $P^s$ hair region mask. The detailed mask design is provided in the Appendix A.1.

**Style Proxy.** Given a style reference image $I_{\mathrm{r}}^{\mathrm{s}}$ or a text prompt $T^{\mathrm{s}}$ of hairstyle. Our goal is to obtain the style proxy $P^s$ to inpaint the source image $I_{\mathrm{src}}$ hair region with the desired hairstyle. We use the hair-agnostic mask $M_{\mathrm{a}}$ to define the inpainting region of the hair. To ensure that the generated hairstyle aligns with the orientation of the face in $I_{\mathrm{src}}$, we employ pose key points image $I_{\mathrm{key}}$ driven from $I_{\mathrm{src}}$ using a 3D key points extractor model $E_p$ to guide the hairstyle inpainting process. The process can be formulated as:

$$P^s = \mathcal{D}(\epsilon_\theta(\mathcal{E}(I_{\mathrm{src}}^{\mathrm{s}} \odot M_{\mathrm{a}}), t, C((E_p(I_{\mathrm{src}}^{\mathrm{s}}), M_{\mathrm{a}})), \psi), \tag{4}$$

where $\psi$ represents the attention conditioning input $[T_E(T^{\mathrm{s}}), I_E(I_{\mathrm{r}}^{\mathrm{s}})]$, $C$ represents the control input from ControlNet.

**Color Proxy.** Given a color reference image $I_r^c$ or a text prompt $T^c$ of hair color. We use the warping module $\mathcal{W}$ to align the $I_r^c$ with $I_c$. The process to get the color proxy operates through the following process:

$$P^c = \mathcal{W}(I_r^c, I_c) \odot M_c^h. \tag{5}$$

Where $M_c^h$ represents the hair region mask of $I_c$. The entire framework can be described by:

$$I_f = \text{HairDiffusion}(I_{\text{src}}, [M_a, M_c], [T^s \cdot T^c \cdot I_r^s \cdot I_r^c]), \tag{6}$$

where $[\cdot]$ denotes optional conditional inputs.

### 3.3  Multi-Stage Hairstyle Blend

We employ MHB to achieve separate control of hairstyle and hair color. Given an input image $I_{\text{src}}$ and the corresponding inpainting mask $M_c$ for the hair region, we obtain the masked image $I_M$, removing the original hair information. We then encode the color proxy $P^c$ and $I_M$ to the latent space of the diffusion model through an encoder $\mathcal{E}$, obtaining the latent vectors $z^c$ and $z^m$ respectively. We use the hyperparameter $\tau$ to divide the denoising process into two stages. In the initial stage, we spatially blend $z^c$ with $z^m$ to obtain modified latent $z_t'$ at a certain intermediate timestep $\tau$ during the sampling, the blending operation $B$ is formulated as follows:

$$z_t' = B(z_t^c, z_t^m, t) = \begin{cases} z_t^c \odot m_c + z_t^m \odot (1 - m_c), & \text{if } t = \tau \\ z_t^m, & \text{otherwise}, \end{cases} \tag{7}$$

where $z_t^c$ is noised color latent at time $t$ and $m_c$ is down sampled from the inpainting mask $M_c$. Owing the blend is in the early sample stage, thereby utilizing $P^c$ to guide hair color generation and avoiding artifacts at the hair boundaries caused by the latent mask mixing. To prevent affecting irrelevant features during denoising, blending $z^m$ ensures that parts outside $m_c$ remain unchanged. Alternatively, hair color can be edited using a textual description $T^c$ by guiding the denoising steps within the U-Net, with text-encoded information via CLIP's text encoder. In the latter stage, we leverage the context-aware capability of inpainting to unconditionally generate the parts of the face and background occluded by the hair. Throughout the entire denoising process, we incorporate $P^s$ or $I_{\text{src}}$ via ControlNet to guide the hairstyle generation.

### 3.4  Hair Color Aligning

To transform the $I_r^c$ into color proxy $P^c$ for hair color editing, we introduce a warping module inspired by the virtual try-on task [5, 22], which is being employed in the domain of hair editing for the first time to best of our knowledge. We adopted the network architecture of HR-VITON [22] incorporates DensePose $D$ [10] and hair-agnostic segmentation map $S_a$ driven from facial image $I_f$ as priors to enable warping module $\mathcal{W}$ to account for facial poses. However, there are two main gaps between the two tasks. Virtual try-on datasets typically consist of two types of images: standalone clothing images and compared clothing images worn on models. These images are jointly used to train models so that they can transfer clothing to images of models. Existing facial datasets lack individual representations for hair. To reduce the gap, we first get the semantic segmentation $M$ of facial image $I_f$ using a facial segmentation network, and the hair mask denotes $M_h$, while the hair-agnostic mask is represented as $M_a'$. Then, we obtain the hair representation $I_h = I_f \odot M_h$ and hair-agnostic facial representation, defined as $I_a = I_f \odot M_a'$. We adopt several data augmentation techniques, denoted as $\mathcal{A}$, including flipping, rotation, trigonometric distortion, and scaling, on the hair representation image $I_h$ to break down the connection with the facial image $I_f$. Formally, the condition input for training the warping module is then given by:

$$\mathcal{W}(\mathcal{A}(I_h), S_a, D, I_a) \rightarrow I_f. \tag{8}$$

This strategy enables the warping module to align the hair region of $I_r^c$ to the hair region of the target hairstyle $I_c$, thereby obtaining $I_w$ as shown in Figure 2(c). When large warping or significant differences in hair length cause image tearing, gaps, or incomplete filling of the target hairstyle areas, we leverage PatchMatch [29] to sense surrounding pixels and seamlessly fill the missing hair color, ensuring natural restoration.

The second gap is to extract the color features while filtering out the high-frequency details of the original hair. To address this, we applied bilateral filtering to eliminate texture details from the hair,

which effectively preserved the original hair color and color structure information intact. We denote the bilateral filtering operation as $\mathcal{G}$, and its application to the image can be represented as:

$$P^c = \mathcal{G}_\theta(I_{\mathrm{w}}), \tag{9}$$

where $\theta$ represents the parameters of $\mathcal{G}$.

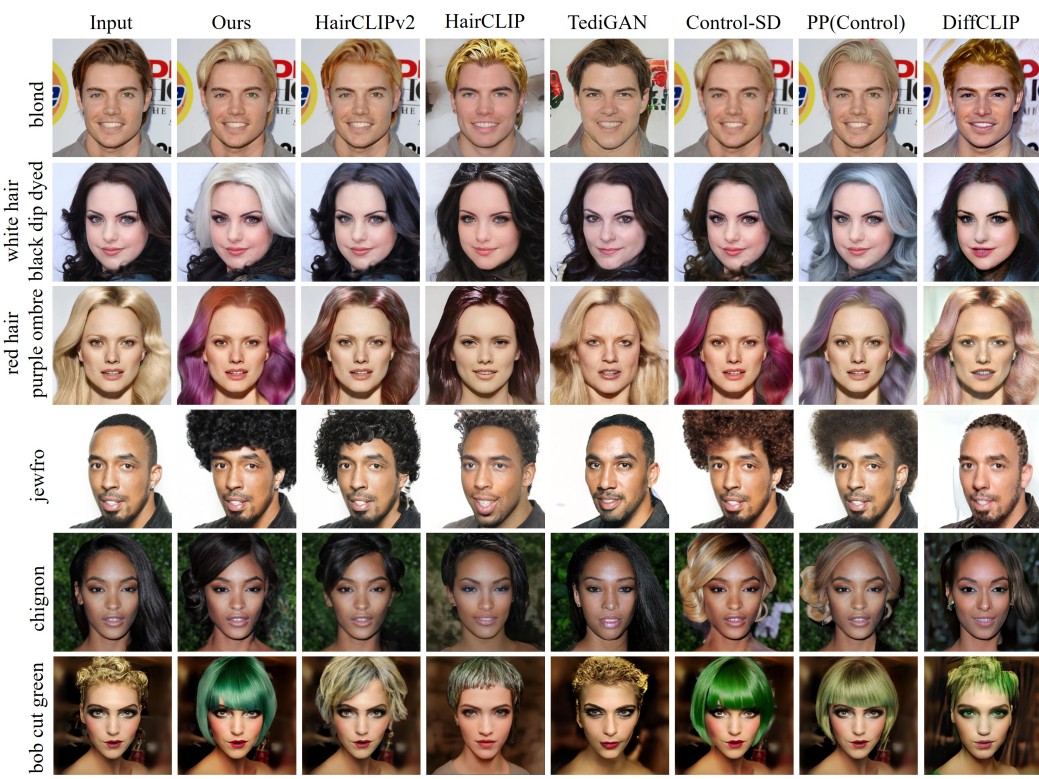

Figure 3: Visual comparison with HairCLIPv2 [36], HairCLIP [35], TediGAN [38], Power-Paint("ControlNet" version) [47], ControlNet-Inpainting [42], and DiffCLIP [18]. The simplified text descriptions (editing hairstyle, hair color, or both of them) are listed on the leftmost side. Our approach demonstrates better editing effects and irrelevant attribute preservation (e.g., identity, background).

## 4 Experiments

**Implementation Details.** We train and evaluate the warping module using a CelebAMask-HQ [21] dataset and obtain paired data corresponding to the hair region through segmentation and transformation processes. For the hairstyle text descriptions and hair color descriptions, we follow the HairCLIP methodology. Additionally, we incorporate additional hair color text descriptions for multi-color commonly used hair color scenarios. The CelebA-HQ [1] dataset is used to provide reference images for hairstyles. Additionally, as the dataset lacks multi-color hair, we supplement it with images sourced from the internet, obtaining multi-hair-color data with a total of 12 categories of hair color structure. Detailed data collection is provided in Appendix A.2. We set the batch size for the training warping module to 8 and trained the module for 100,000 iterations. The learning rates for both the generator and the discriminator in the warping module are set to 0.0002.

### 4.1 Quantitative and Qualitative Comparison.

**Comparison with Text-Driven Hair Editing Methods.** Table 1 shows the quantitative results on IDS, PSNR, and SSIM with the leading text-driven hair editing methods on the CelebA-HQ [1] testset(2,000 images), following the evaluation setting of HairCLIPv2. For Diffusion-CLIP [18], we finetune a model for each text description. For the TediGAN [38], the number of optimization iterations is set to 200. We set the image generation size to 1024×1024, consistent with the compared

methods. Our method accomplishes maximizing the preservation of irrelevant attributes. Figure 3 shows the qualitative results in StyleGAN-based and advanced SD-inpainting methods. The three diffusion-based methods are based on Stable Diffusion v1.5 in 50 steps. As shown in Figure 3, our method accomplishes satisfactory hair editing effects with hair color structure text. In text editing hairstyle scenarios (lines 4 and 5), compared to the diffusion-based method, we can faithfully preserve the original hair color. The StyleGAN-based methods are unsatisfactory in the preservation of irrelevant attributes. Even though HairCLIPv2 performs well visually, it still struggles to preserve details, as shown in Figure 4.

| Methods | IDS↑ | PSNR↑ | SSIM↑ |
|---|---|---|---|
| **Ours** | **0.94** | **33.1** | **0.95** |
| HairCLIPv2 [36] | 0.84 | 29.5 | 0.91 |
| HairCLIP [35] | 0.45 | 21.6 | 0.74 |
| TediGAN [38] | 0.16 | 22.5 | 0.74 |
| DiffCLIP [18] | 0.71 | 26.8 | 0.86 |

Table 1: Quantitative comparison for irrelevant attributes preservation. IDS [13] denotes identity similarity, PSNR, and SSIM are calculated at the intersected non-hair regions before and after editing.

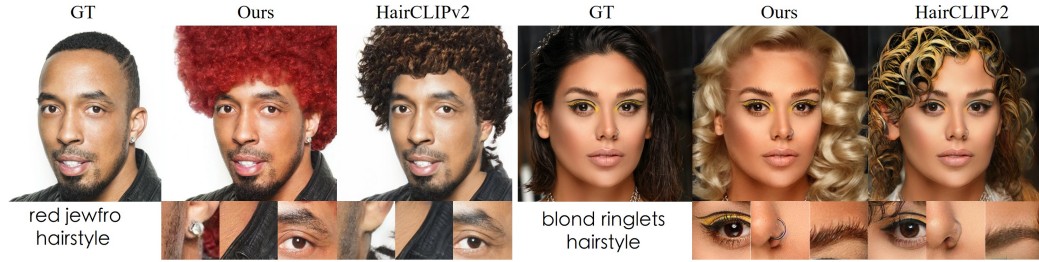

Figure 4: Comparison with HairCLIPv2 [36] in detail. Our approach shows better preservation of irrelevant attributes.

**Comparison with Hair Color Transfer Methods.** Due to the existing facial datasets lack of multi-color data, we randomly selected 666 images from our collected multi-color dataset as color reference images. The first 666 images from the CelebA-HQ [1] test set as input images. As shown in Figure 5, our result excels at transferring color structures while remaining faithful to the hair color of the reference image. Although MichiGAN [33] demonstrate the ability to achieve multiple color structure, however, it results in poor hairstyle quality. On the other hand, HairCLIP, HairCLIPv2, HairFastGAN and Barbershop are constrained by the data distributions of StyleGAN, limiting their ability to generate novel colors within the latent space.

**User Study.** As shown in Table 2, our method outperformed other methods in terms of accuracy. In terms of preservation, our method was superior to traditional hair editing methods but slightly inferior to PowerPaint [47], which is specifically designed for inpainting using diffusion techniques. In the color transfer comparison, due to our hair color feature blending mechanism, when editing hair color via color proxies that significantly differ from the target image, the overall image color balance shifts, resulting in a preservation score lower than MichiGAN [33]. Regarding naturalness, our method was slightly inferior to SYH [19], which performs optimization entirely in the latent space of StyleGAN. The detailed user study setting is provided in Appendix A.3. Visual comparison on cross-model is provided in Appendix A.4.

| | Text-Driven | | | | | | | Color Transfer | | | | | | | Cross-Model | | |
|---|---|---|---|---|---|---|---|---|---|---|---|---|---|---|---|---|---|
| Metrics | Ours | [36] | [35] | [38] | [18] | [42] | [47] | Ours | [36] | [35] | [45] | [19] | [33] | [11] | Ours | [36] | [35] |
| Accuracy | **42.9** | 21.3 | 13.2 | 1.5 | 1.3 | 5.0 | 14.8 | **64.5** | 8.0 | 4.7 | 10.8 | 1.8 | 6.8 | 3.5 | **68.3** | 21.7 | 10.1 |
| Preservation | 24.5 | 20.5 | 2.1 | 1.5 | 3.3 | 23.3 | **25.1** | 21.3 | 20.5 | 2.8 | 10.5 | 16.0 | **23.8** | 5.3 | **48.5** | 38.5 | 13.0 |
| Naturalness | **27.8** | 24.8 | 6.3 | 2.3 | 0.3 | 21.5 | 26.3 | 26.3 | 7.3 | 9.8 | 3.8 | **28.3** | 2.3 | 22.3 | **55.3** | 36.8 | 8.0 |

Table 2: User study on text-driven image manipulation, color transfer, and cross-modal hair editing methods. Accuracy denotes the manipulation accuracy for given conditional inputs, Preservation indicates the ability to preserve irrelevant regions and Naturalness denotes the visual realism of the manipulated image. The numbers represent the percentage of votes. **Bold** and underline denote the best and the second best result, respectively.

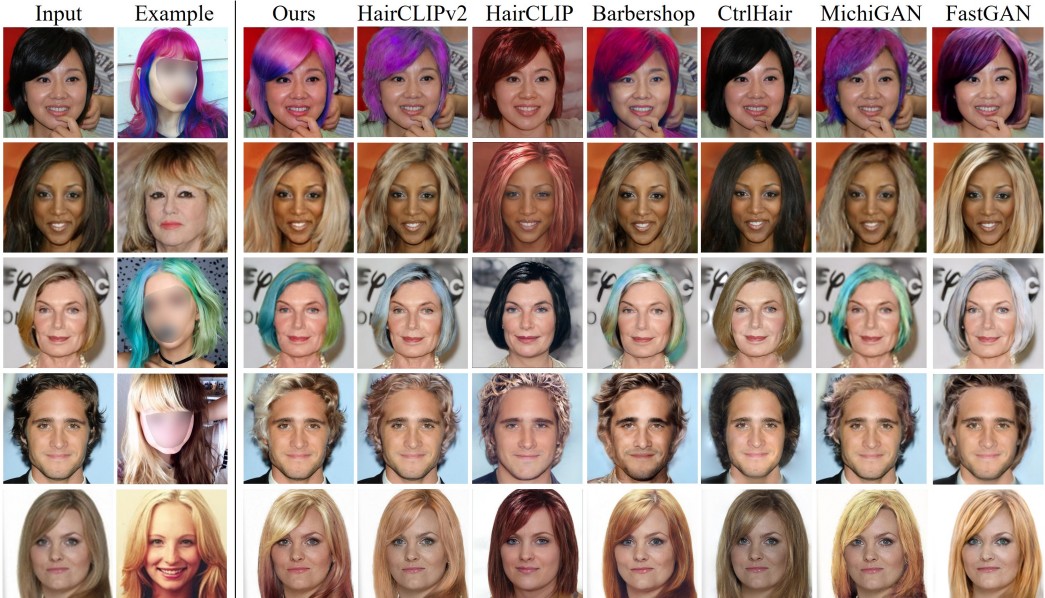

Figure 5: Visual comparison with HairCLIPv2 [36], HairCLIP [35], Barbershop [45], CtrlHair [11], MichiGAN [33] and HairFastGAN [26] on hair color transfer.

## 4.2 Ablation Study

To demonstrate the effect of each step in our pipeline, we conducted ablation experiments by incrementally adding conditions. As shown in Figure 6, the first row illustrates hairstyle editing based on a given hairstyle text. In the original image, the absence of an agnostic mask results in the failure to generate a bob-cut hairstyle with bangs, leading to poor hairstyle generation quality. In the second step, the lack of an Openpose ControlNet [42]causes misalignment between the generated hairstyle and the face. The third step, without a color proxy, results in uncontrollable hair color. In the fourth step, the introduction of an unwarped color proxy causes random color generation in regions lacking hair color guidance from the original image and alters the background color due to the color proxy's influence. The fifth step aligns the color proxy with the target hairstyle area, ensuring the hair color matches the Source Image. In the second column, a reference hair color image is used for hair color editing. In the original image, the absence of a Canny edge ControlNet results in an uncontrollable hairstyle. In the third image, the Canny edge ControlNet controls the hairstyle structure but not the color. In the fourth image, using the reference image directly as a stroke map for the color proxy results in the region without hair in the reference image lacking color, with the hair color also appearing in the background. In the fifth image, the absence of Bilateral Filtering in the hairstyle causes the warped hairstyle's structural features to adversely affect the original hairstyle structure, leading to poor hairstyle structure.

To better assess the individual contribution of the warping module, we observed that after performing the warp operation, significant discrepancies in hairstyle length or complex edges can lead to alignment challenges, as illustrated in Figure 7. However, by incorporating the patch match method for inpainting, we can effectively reconstruct the corresponding hair color. Additionally, the removal of texture information through blurring further enhances the results. The influence of each configuration on hairstyle generation quality is presented in Table 3.

| Model | FID↓ | FID$_{\text{CLIP}}$↓ | SSIM↑ |
|---|---|---|---|
| w/o warping module | 33.17 | 12.53 | 0.62 |
| w/o patch match | 27.74 | 8.51 | 0.70 |
| w/o bilateral filtering | 20.85 | 6.02 | 0.74 |
| HairDiffusion | **20.83** | **5.96** | **0.76** |

Table 3: Quantitative comparison of different variants of warping module with various conditions removed. We achieve the best performance by leveraging the remaining techniques.

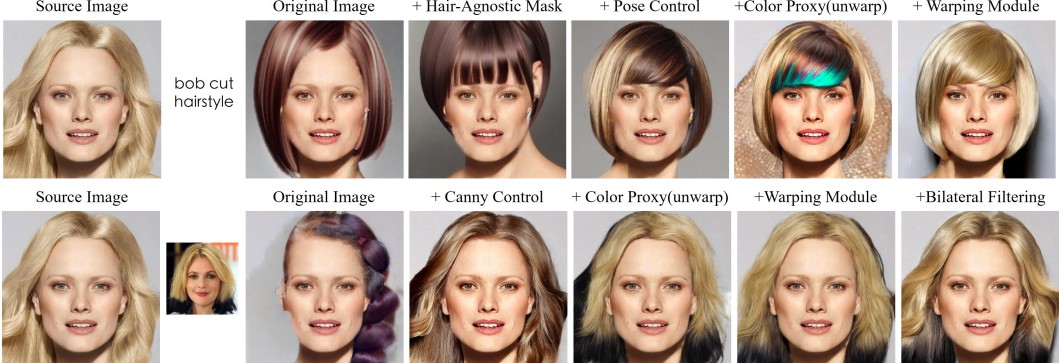

Figure 6: Ablation studies on text-guided hairstyle editing and reference image-guided hair color editing.

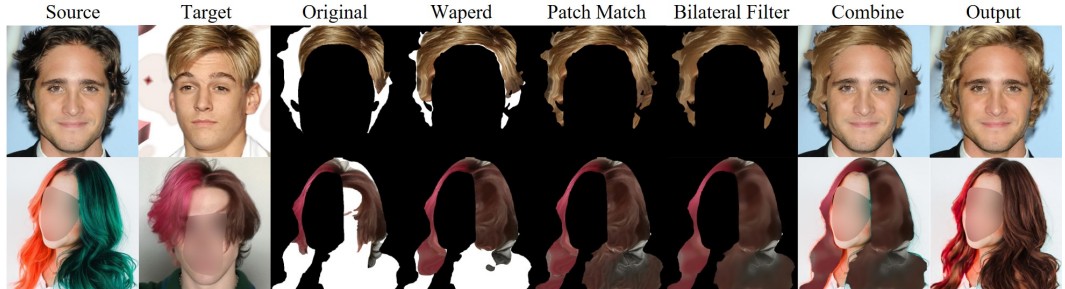

Figure 7: Visualizations of the ablation studies on the warping module and corresponding post-processing.

## 5 Conclusion

In this work, we first propose the latent diffusion-based approach for hair editing. We introduce the MHB module and hair-agnostic masks, which enable the diffusion model to effectively control hairstyle and hair color independently while preserving unrelated attributes. Additionally, we employ a warping module for the first time in this task to ensure alignment of hair color, demonstrating its capability in hair color manipulation and preservation. Furthermore, by collecting image-text pairs focused on hair color structure, we further enhance our model's ability to finely control hair color using both text and reference images.

## 6 Acknowledgements

This work is supported by the National Natural Science Foundation of China under Grant 62176163, 82261138629, and 62320106007, the Science and Technology Foundation of Shenzhen under Grant JCYJ20210324094602007, the Guangdong Basic and Applied Basic Research Foundation under Grant 2023A1515010688, and the Guangdong Provincial Key Laboratory under Grant 2023B1212060076, and the Shenzhen Higher Education Stable Support Program General Project under Grant 20231120175215001.

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

# A    Appendix / supplemental material

## A.1    Hair-Agnostic Masks Design

To transform the hair editing task into an inpainting task, we utilize hair-agnostic masks designed for hair editing tasks at different stages, as shown in Figure 8.

**Hairstyle Editing stage.** As depicted in Figure 2(a), starting with a source image $I_{\text{src}}$, we obtain the hair-agnostic region mask $M_a$ to transform the task into an inpainting task. Aiming to obtain a satisfactory style proxy $P^s$, we focus on retaining facial and neck information while removing other irrelevant details (background, hair, etc). Additionally, for hairstyles with bangs, forehead information is also removed as part of this process. To get the $M_a$, The first step is to select reference images, (automatically) segment them, and select regions in the reference images that should be copied to the target image. Let $M_I = \text{SEGMENT}(I_{\text{src}})$ indicate the segmentation of reference image $I_{\text{src}}$, where SEGMENT is a facial semantic segmentation network such as BiSeNET [41]. The hair region of it is $M_I^{\text{h}}$. For the same, we also segment the $I_c$ to get segmentation $M_c$, and the hair region of it represents $M_c^{\text{h}}$. To ensure effective bangs generation, we obtained a series of key point coordinates using a facial key point detection model [4]. These key points are represented as $K : (k_i^x, k_i^y)$, where $i = 1, 2, \ldots, n(n = 68)$ denotes the index of the key points. Assuming that the key points within the eyebrow region are $(k_b^x, k_b^y)$, for $b = n, n+1, ..., m(m - n = 9)$, we employed a Bézier curve fitting approach to identify a dividing line beneath the eyebrows, splitting the face into two parts. Let $B(t)$ represent the equation of this curve. For any pixel $(x, y)$ located on $M_I^1$, if it falls below the curve defined by $B(t)$, it is classified as part of the area $M_I^3$ below the forehead. To preserve other facial attributes such as ears and neck, we superimpose relevant attribute labels segmentation $M_2$. The final hair-agnostic mask can be represented as:

$$M_a = M_I^2 \cup M_I^3. \tag{10}$$

**Hair Color Editing stage.** Illustrated in Figure 2(b). If editing the hairstyle, the hair region of style proxy $P^s$ and the target image $I_{\text{src}}$ are not aligned, the purpose of the agnostic mask $M_c$ in this stage serves two functions: 1)It removes the hair information from $I_{\text{src}}$ while preserving information outside of the hair area, such as the background, face, and neck. 2)It prepares the mask to be suitable for editing with the $P^s$. This involves removing information in $I_{\text{src}}$ corresponding to the hair region in the style proxy to ensure effective hairstyle editing. So in this stage, the hair-agnostic mask can be represented as:

$$M_c = M_s^{\text{h}} \cup M_I^{\text{h}}. \tag{11}$$

Notably, the region $M_n$, defined as $M_n = M_I^{\text{h}} \cap \neg M_s^{\text{h}}$, can be inpainted using the context-aware capabilities of stable diffusion to achieve a reasonable background restoration.

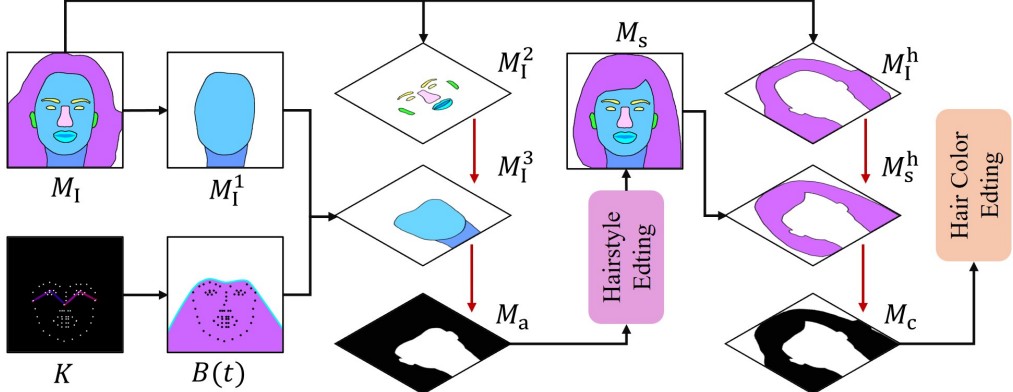

Figure 8: Utilizing facial semantic segmentation and facial key points, we generate the hair-agnostic masks.

## A.2    Data Collection

We crawled tens of thousands of multi-color hair images using keywordsand conducted a thorough data cleaning process, resulting in approximately 6,000 valid images. The cleaning process involved

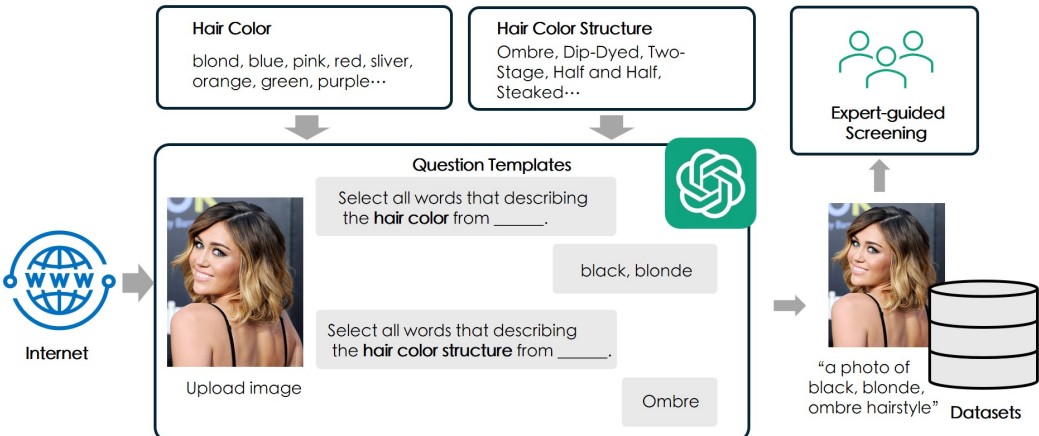

Figure 9: Pipeline for collecting multi-color hairstyle data and corresponding text annotations.

removing duplicates, low-resolution or blurry images, and filtering out irrelevant content (e.g., images without hair or those not fitting the multi-color criteria). We also manually and automatically excluded images with watermarks or copyright markings. To ensure privacy and ethical use, we applied blurring techniques to remove facial information since only hair information was necessary. These images were then input into GPT-4 [2] with predefined hair color and structure categories to generate text annotations like "a photo of {colors}, {color structure} hairstyle." We employed 10 professional annotators who rated the annotations through multiple rounds, discarding low-rated images and retaining 4,625 high-quality multi-color text-image pairs. This dataset was subsequently used to fine-tune the text encoder of the CLIP model, enhancing its performance in multimodal tasks.

### A.3 User Study Setting

For the above three comparisons, we recruited 20 volunteers with backgrounds in computer vision to conduct a comprehensive user study. We randomly selected 20 sets of results from each experiment, forming a total of 60 test samples. During the testing process, the order of different methods was randomly shuffled. For each test sample, volunteers were asked to choose the best option.

### A.4 Comparison with Cross-Modal Hair Editing Methods.

| Input Image | Ours | HairCLIPv2 | HairCLIP | Ours | HairCLIPv2 | HairCLIP |

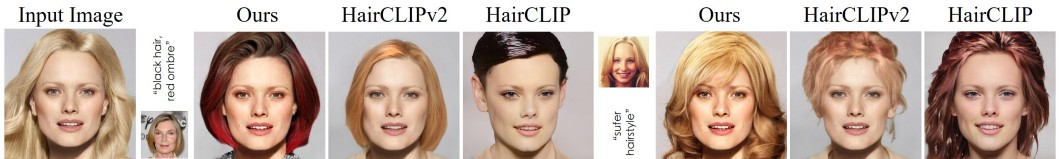

Figure 10: Qualitative comparison with HairCLIP and HairCLIPv2 on cross-modal conditional input. Our approach shows better editing effects and excellent preservation of irrelevant attributes.

### A.5 Examples of Reconstruction

We have supplemented self-transfer experiments with state-of-the-art methods HairFastGAN [26] and StyleGAN Salon [17]. The comparison illustrates the retention of accessories (D, G, H, J, M, O); multi-color hair preservation (N, Q); hand preservation (F, K, L, U, W); unique facial features retention (I); hairstyle preservation (A, B, C, E, T, S, R, V); background retention (P). Our method also has limitations, such as color discrepancies in attributes other than the hairstyle in some cases, like hair color in the second column and skin color in the seventh column. This is due to merging the noisy latent, masked image latent, stroke map latent, and mask at the initial convolution layer of the UNet architecture, where they are collectively influenced by the text embedding. Consequently, subsequent layers in the UNet model struggle to obtain pure masked image features due to the influence of the text and stroke map. Quantitative comparisons of the reconstructions are shown on the right side of the Table 4.

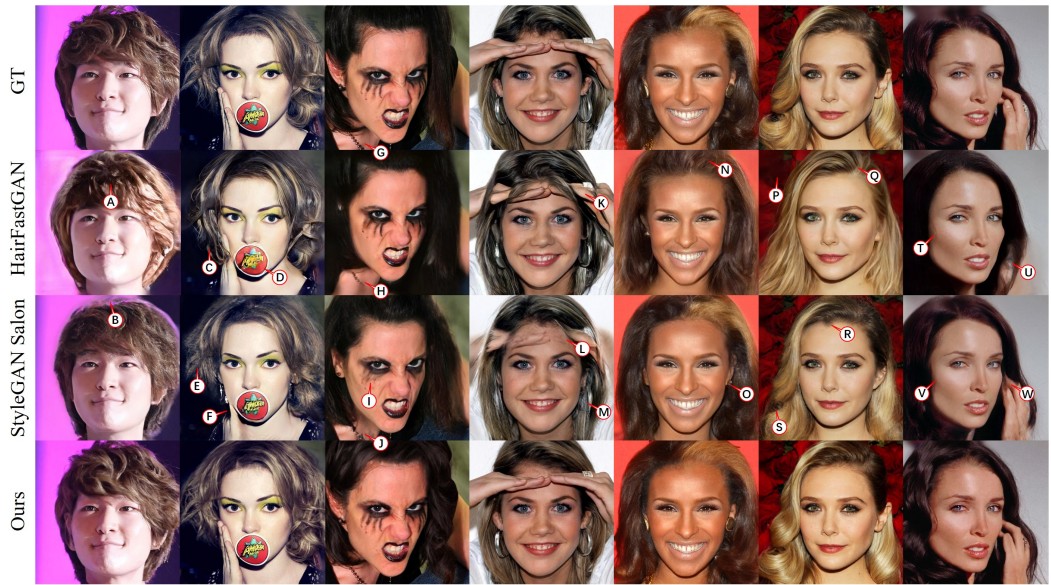

Figure 11: Visualization of the reconstruction comparison with StyleGAN Salon and HairFastGAN.

## A.6 Limitations

Our method also has multiple limitations while we have trained a warping network coupled with inpainting to align the target hair color with reference images, achieving direct color transfer remains challenging for reference hairstyles that differ significantly in structure or facial pose. For instance, with styles like ombre hairstyle, it becomes difficult to produce plausible outcomes if the reference image features short hair while the target image features long hair. Similarly, when only a profile view is available, transferring hair color to a frontal view becomes problematic.

Given the method's reliance on segmentation networks, its performance is constrained by the segmentation network's ability to accurately delineate facial regions, especially the hair area. For intricate styles like cornrows hairstyle, which pose challenges for segmentation, achieving satisfactory results becomes notably difficult. In the task of single-color transfer, as shown in Table 4, the metrics for this method are not satisfactory. However, this can be addressed by utilizing more advanced diffusion models.

**More Examples of the Limitations.** Figure 15 shows the limitations of the warping module in extreme cases of hair color transfer, including significant pose differences, complex textures, and large discrepancies in hairstyle regions. To visually illustrate the limitations of the warping module method, the warped results have not undergone the post-processing mentioned in the paper. The first and second rows demonstrate cases with significant differences in hair length. While the hairline region aligns well, the hair ends do not; The second row on the left, where the hair lengths are similar, performs well. This is due to the consistent inclusion of hair end information when generating the paired hair dataset used to train the warping module. The third, fourth, and fifth rows showcase significant pose differences. The warped result on the right side of the second row shows that the hair orientation aligns well with the facial orientation of the Source Image, indicating some robustness in the model across different poses. However, the right side of the second row and the third row still show misalignment in hair parting, leading to color inconsistencies in the output. The left side of the fourth row shows the model discarding the hair ends while supplementing the bangs area of the Source Image, though there is still some deviation in the hair color's centerline. The last row depicts complex texture scenarios where the Output hair color does not match the Target Image, due to the compression of images required for diffusion input and the bilateral filtering operation removing high-frequency color details. For cases with missing target region hair color, post-processing with patch match can fill in the blank areas, as shown in Figure 7.

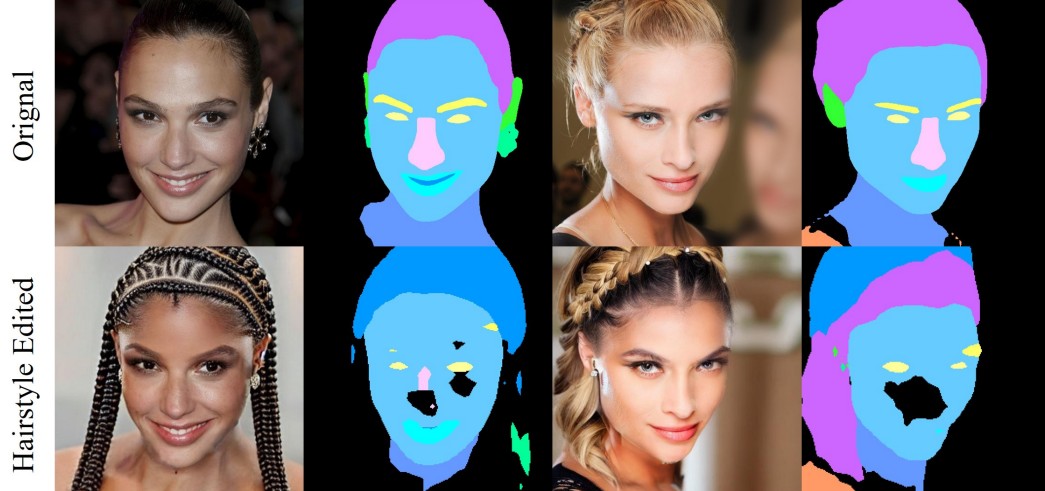

Figure 12: For complex hairstyles, existing face segmentation models struggle to accurately identify intricate hair patterns. Additionally, even though measures are taken to preserve the face and background during the multiple stages of image editing, the segmentation map indicates that there might be issues with correctly segmenting the facial regions.

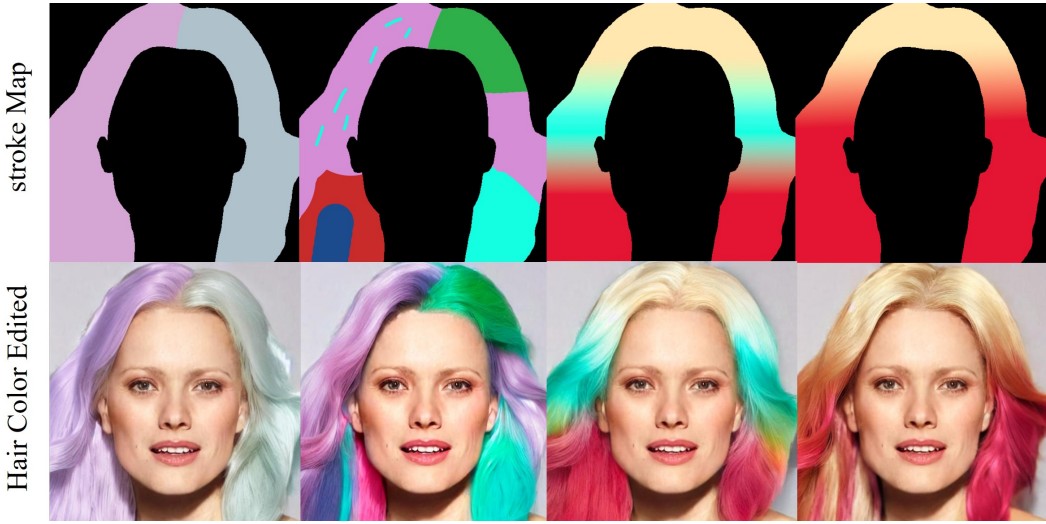

Figure 13: By using a stroke map encoder in the latent space, it inevitably overlooks details of hair color, as shown in the second column. And it does not completely match the provided hair color areas, as shown in the fourth column.

## A.7 Social Impact

**Negative Impact.** Hair editing tasks can also carry some negative implications and potential risks: 1) Social Bias: Similar to image inpainting models, hair editing models rely on internet-collected training data, which may contain social biases. Certain hairstyles could be incorrectly associated with specific races or genders, reinforcing existing social stereotypes. 2) Privacy Concerns: Hair editing involves modifying personal photos, which, if done without consent, can infringe on individual privacy. 3) Misleading Information: The technology can be used to alter photos misleadingly, potentially spreading false information. This can be used maliciously to modify images of public figures or ordinary people, leading to misinformation or cyberbullying. 4) Aesthetic and Identity Issues: Hairstyles are an important part of personal identity. Altering someone's hairstyle in a photo without their consent can negatively affect their self-image and identity. To address these concerns, it is crucial to emphasize responsible use and establish ethical guidelines when utilizing hair editing technology. This is also a key focus for our future model releases.

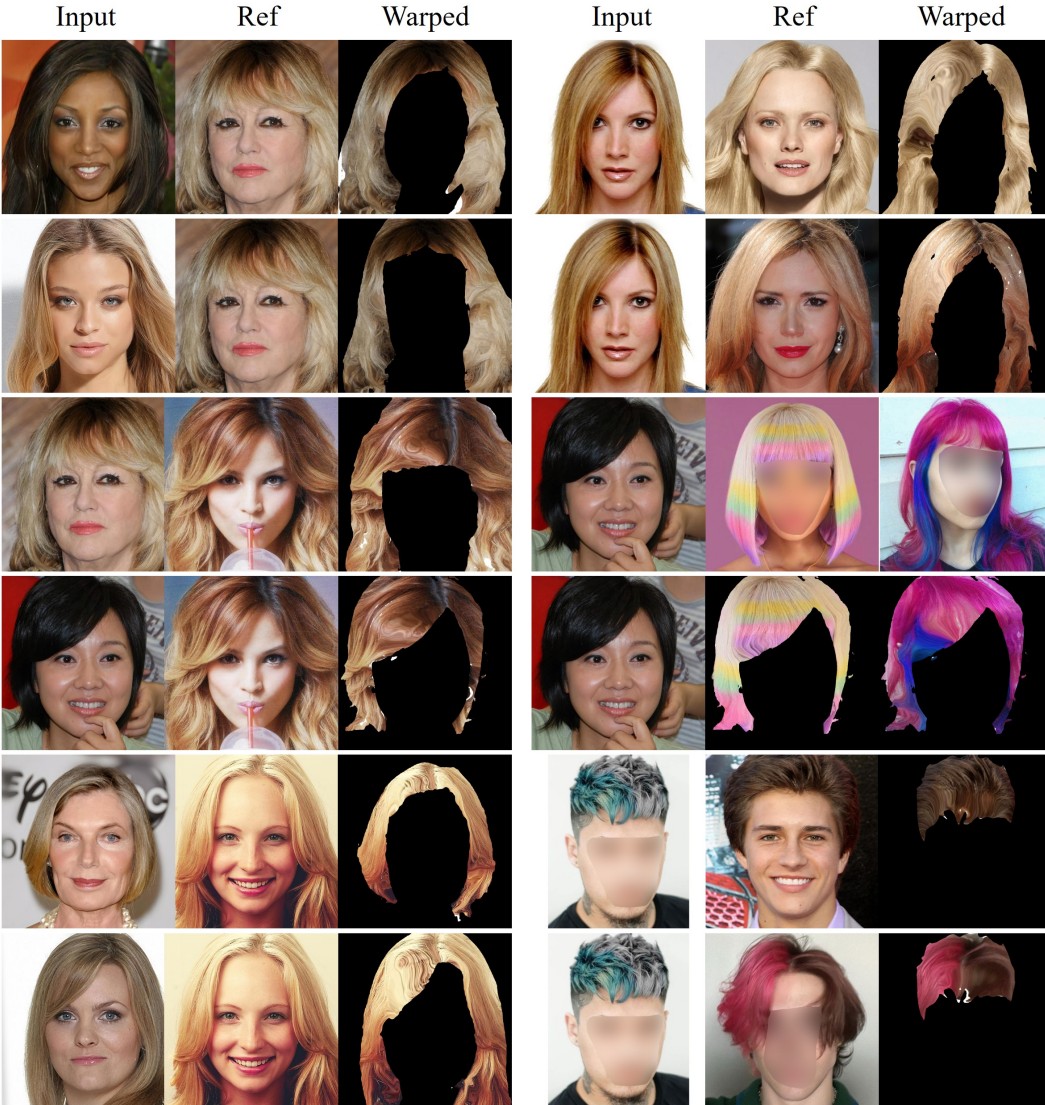

|    Input    |    Ref    |    Warped    |    Input    |    Ref    |    Warped    |

Figure 14: The result of the warping module, ensures alignment of hair color with the target hair region, demonstrating its capability in hair color manipulation and preservation. To demonstrate the direct effect of the warping model, this shows the results without bilateral filtering processing. In cases where the hairstyles differ significantly, some areas may appear empty.

**Positive Impact.** Hair editing tasks also offer several positive impacts: 1) Creative Expression: Hair editing allows users to experiment with different hairstyles and colors, promoting creativity and self-expression. It enables people to visualize new looks before making real-life changes. 2) Fashion and Beauty Industry: Professionals in the fashion and beauty industry can use hair editing tools for styling consultations, marketing campaigns, and virtual makeovers, enhancing customer engagement and satisfaction. 3) Accessibility: Hair editing technology can help individuals who may not have access to professional styling services to explore and enjoy different hairstyles virtually.

## A.8 Detailed information

We perform the inference of different diffusion-based methods in the same setting unless specifically clarified, i.e., on NVIDIA A40 following their opensource code with a base model of Stabe Diffusion v1.5 in 50 steps, with a guidance scale of 7.5. We use an Adam optimizer with a batch size of 16. The initialized learning rate for the generator is set to 0.0002 and for the discriminator is also set to 0.0002.

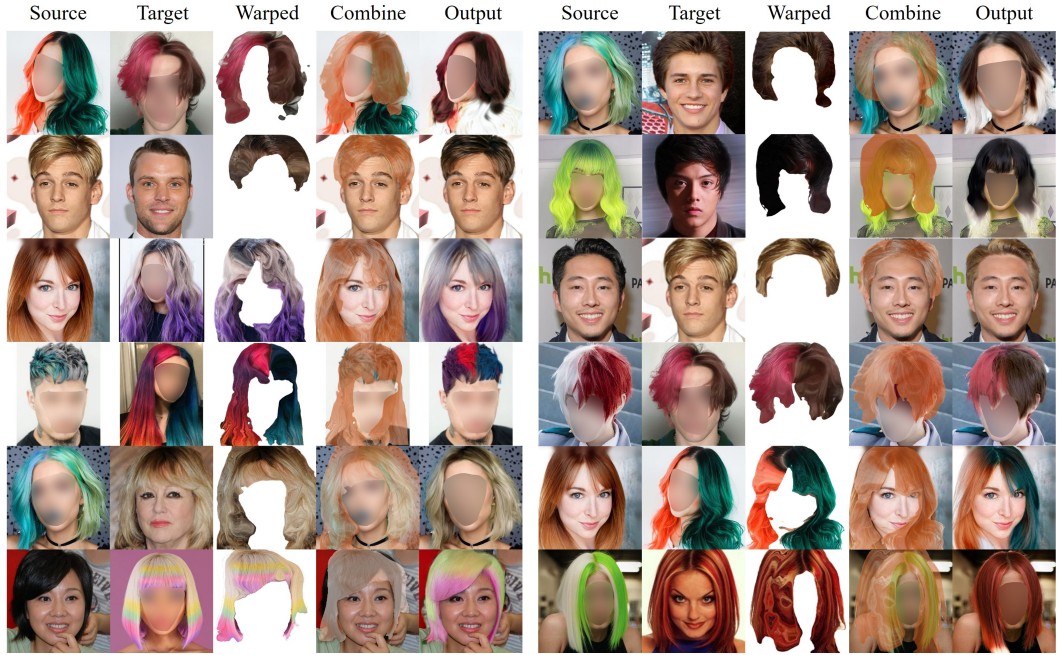

| Source | Target | Warped | Combine | Output | Source | Target | Warped | Combine | Output |

Figure 15: Extreme Cases of the warping module, warped is the unprocessed output of the warping module. Combine is a visualization of the alignment between the warped output and the corresponding region of the ground truth image. Output treats the warped image as a color proxy for the source image.

| Model | Single Color transfer | | Reconstruction | | | |
|---|---|---|---|---|---|---|
| | FID↓ | FID$_{CLIP}$↓ | LPIPS↓ | PSNR↑ | FID↓ | FID$_{CLIP}$↓ |
| HairCLIP | 40.08 | 10.94 | 0.36 | 14.08 | 35.49 | 10.48 |
| HairCLIPv2 | 20.21 | 6.55 | 0.16 | 19.71 | 10.09 | 4.08 |
| CtrlHair | **19.65** | 3.62 | 0.15 | 19.96 | 8.03 | 1.25 |
| StyleYourHair | - | - | 0.14 | 21.74 | 10.69 | 2.73 |
| Barbershop | 20.54 | 3.89 | 0.11 | 21.18 | 13.37 | 2.61 |
| HairFastGAN | 20.17 | **3.00** | 0.08 | 23.45 | 9.72 | 0.97 |
| HairDiffusion | 20.83 | 5.96 | **0.07** | **31.66** | **5.41** | **0.68** |

Table 4: Quantitative comparison of single color transfer and self-transfer reconstruction metrics. Bold and underline denote the best and the second best result, respectively.

