# OpenReview forum: "HairDiffusion: Vivid Multi-Colored Hair Editing via Latent Diffusion"
_NeurIPS.cc/2024/Conference — NeurIPS 2024 poster_

### Official Review · Reviewer_GkwL · 2024-07-04

**Soundness:** 3
**Presentation:** 2
**Contribution:** 3
**Rating:** 6
**Confidence:** 2

**Summary:**

### NeurIPS Review

#### Summary:
This submission presents an innovative 2D hairstyle editing pipeline leveraging latent diffusion models (LDM). The key component of this method is the Multi-Stage Hairstyle Blend (MHB), which facilitates separate control over hairstyle and hair color. By integrating structural information from non-hair regions such as facial masks and keypoints, the method effectively preserves non-hair attributes in the editing results. Additionally, a hair warping module enables natural transfer of hair color from the source image to the generated target image. Extensive experiments on public datasets like CelebA-HQ demonstrate improved results compared to previous state-of-the-art methods.

#### Strengths:


#### Weaknesses:

**Strengths:**

- Clarity and Writing: The paper is well-written and relatively easy to understand.
- Technical Contributions: The submission presents a comprehensive system with solid and relatively novel technical contributions.
      * Hair Warping Module: The inclusion of a warping module that transfers the color pattern from the source hairstyle to the target hairstyle is notable.
      * Multi-Stage Hairstyle Blend (MHB) Module:** This module is effective in preserving non-hair attributes in the generated image.
- Extensive Experiments:
  - The method produces more natural hair editing results compared to previous approaches, maintaining consistent hair color with the input text/image (control source) and better preserving non-hair regions.
  - A user study validates the human preference for the presented method over other methods.
  - The ablation study includes visualizations demonstrating the unique contributions of each module/control signal to the final results, along with detailed discussions.

**Weaknesses:**

- More discussions on the failure cases: Most results are shown with a near-frontal head pose. Including and discussing more results with non-frontal head poses would enhance the completeness and understanding of the work. Additionally, as mentioned in the limitation section, the presented method might not work well with the color transfer between different hairstyles. It would also be great if more failure cases are included on that end.
- Manuscript Quality: There are some typos and missing text. A thorough proofreading is necessary to refine the manuscript.

**Questions:**

See weakness.

**Limitations:**

Yes.

---

> ### Author Rebuttal · Authors · 2024-08-06
>
> Thank you for your suggestions. We have included additional failure cases in Figure 1 of the PDF, where we comprehensively demonstrate the effects of multi-view hair color transfer and discuss the reasons for poor performance.
>
> In Figure 1 of the PDF, we have added discussions on the limitations of the warping module in extreme cases of hair color transfer, including significant pose differences, complex textures, and large discrepancies in hairstyle regions. To visually illustrate the limitations of the warping module method, the warped results in the figure have not undergone the post-processing mentioned in the paper.
> The first and second rows demonstrate cases with significant differences in hair length. While the hairline region aligns well, the hair ends do not; however。the second row on the left, with similar hair lengths, performs well, which is related to not randomly removing hair end information when generating the paired hair dataset. The third, fourth, and fifth rows show cases with significant pose differences. The warped result on the right side of the second row shows that the hair orientation aligns well with the facial orientation of the Source Image, indicating some robustness in the model across different poses. However, the right side of the second row and the third row still show misalignment in hair parting, leading to color inconsistencies in the Output. The left side of the fourth row shows the model discarding the hair ends while supplementing the bangs area of the Source Image, though there is still some deviation in the hair color's centerline. The last row depicts  complex textures scenarios where the Output hair color does not match the Target Image, due to the compression of images required for diffusion input and the bilateral filtering operation removing high-frequency color details.
> For cases with missing target region hair color, post-processing with PatchMatch can fill in the blank areas, as shown in Figure 4. The effectiveness is demonstrated in the ablation experiments in Table 1.
>
> We will correct typos and missing text in the future version. Thank you for your assistance!

---

> ### Author Response · Authors · 2024-08-13
>
> Due to the upcoming deadline for the discussion, I would like to confirm if you have any further questions or if there are areas where you need further clarification. If there are no more issues with my submission, I hope to receive your feedback and proceed.
>
> Thank you very much for taking the time to review my work amidst your busy schedule. I greatly value your opinions and hope to improve my research under your guidance.
>
> Thank you for your assistance, and I look forward to your reply.

---

### Official Review · Reviewer_X6RQ · 2024-07-11

**Soundness:** 3
**Presentation:** 2
**Contribution:** 3
**Rating:** 4
**Confidence:** 3

**Summary:**

This paper presents a new framework for hair editing tasks, which includes editing hair color and hairstyle using text descriptions, reference images, and stroke maps. The proposed approach leverages Latent Diffusion Models (LDMs) and introduces the Multi-stage Hairstyle Blend (MHB) technique to effectively separate the control of hair color and hairstyle. Additionally, the method incorporates a warping module to align hair color with the target region, enhancing multi-color hairstyle editing. The approach is evaluated through extensive experiments and user studies, demonstrating its superiority in editing multi-color hairstyles while preserving facial attributes.

**Strengths:**

The proposed method demonstrates impressive performance in multi-colored hair editing, showcasing the ability to handle complex hair color structures while preserving facial attributes effectively. The integration of Latent Diffusion Models (LDMs) and the Multi-stage Hairstyle Blend (MHB) technique provides a novel approach to decoupling hair color and hairstyle, enhancing the quality of the edited images.

**Weaknesses:**

1. The overall pipeline primarily consists of two stages:
   Altering the Hairstyle: This involves using a combination of control net and diffusion models with a hair-agnostic mask, 2D body pose keypoints, prompts, and a reference image as conditions. These modules are commonly found in previous methods.
     Editing the Hair Color: This stage blends information between the hair mask and source image, along with the Canny image of the stylized image and a prompt, to achieve the final result. The main contribution lies in the Multi-stage Hairstyle Blend (MHB) method, which warps the reference hair to the source image to better maintain the hairstyle. However, warping modules are also commonly used in previous methods [4, 21].
  The overall technical contribution does not meet the bar for this conference.

2. The hair structure of the source image is not well-maintained during color editing. As demonstrated in Fig. 5, fourth row, the strand direction of the hairline in the generated image is different from the input image. The results of HairClipv2 better preserve the curliness structure of the original input hair.  Additionally, the generated image in the third row of Fig. 5 shows noticeable brightness differences and a distinct color discrepancy compared to the input image.

3. The paper's writing lacks clarity. It would be beneficial to specify the difference between \( I_c \) and \( I_i \) in Fig. 2, within the caption of this figure. Additionally, using \( I_i \) to denote the style proxy creates confusion.

**Questions:**

Please check Weaknesses.

---

> ### Author Rebuttal · Authors · 2024-08-07
>
> Thank you to the reviewers for pointing out the weaknesses.
>
> **W1:** This study is the first to propose a diffusion-based pipeline in the hair editing field, addressing the issue of **multi-color hair structure** in **text2img** and **img2img** scenarios. Previously, no methods specifically focused on the transfer of multi-color hairstyles, and our visual effects, whether in text2img or img2img, are among the most effective to date. We introduced the Warping module into the hair editing domain to align hair colors, resulting in a Color Proxy that enables the model to faithfully align the target hair color while maintaining the hairstyle structure. To achieve color alignment through the warping module, we undertook the following work:
> 1. Addressing **the lack of paired datasets**: we used semantic segmentation and data augmentation to obtain the hair region deviating from the original face.
> 2. **Preserving hairstyle structure** by removing low-frequency details: bilateral filtering was employed to remove low-frequency details, enhancing the quality of hairstyle generation. Overall, the Warping Module can provide guidance for future researchers applying it in the hair editing field.
> Our proposed hair-agnostic mask is also crucial for enabling hair editing tasks using diffusion methods, as it maximizes the retention of hairstyle-independent attributes while preserving the editability of the hairstyle region. Although our pipeline incorporates components from existing technologies, we focus on integrating and optimizing these techniques to address specific issues in practical applications. For instance, how to blend hair color and hairstyle to achieve balance while reducing artifacts, filling blank areas without a hairstyle, and improving the quality of generated hairstyles. Additionally, how to achieve more flexible editing by decoupling hair color and hairstyle through multiple steps are all considerations for practical application.
>
> **W2:1. Color discrepancy in facial images**
>
> The issue of color discrepancy outside the inpainting region is a common challenge in diffusion-based inpainting tasks. For example, in the ControlNet-Inpainting model used in our paper, the initial convolution layer of the UNet architecture merges the noisy latent, masked image latent, and mask, all of which are collectively influenced by the text embedding. Consequently, subsequent layers in the UNet model struggle to obtain pure masked image features due to the text's influence. The paper "BrushNet: A Plug-and-Play Image Inpainting Model with Decomposed Dual-Branch Diffusion" proposed a blending operation to address this issue. It presents a simple pixel space solution by first blurring the mask and then performing copy-and-paste using the blurred mask. It is important to emphasize that this is a common challenge for diffusion inpainting and is beyond the scope of this paper.
>
> **2. Preserving hairstyle details**
>
> There are indeed slight changes in the generated hairstyle structure compared to the original, but the overall structural control is good. HairCLIPv2 does not faithfully reproduce the target image's hair color at all. This paper primarily focuses on multi-color transfer. ControlNet, as a sparse control matrix, relies on canny maps for hairstyle structure retention. However, Canny maps cannot achieve pixel-level control, resulting in minor differences between the generated and original hairstyles. These differences do not affect the overall hairstyle structure. Training a more refined canny map with a hairstyle dataset could potentially improve the quality of hairstyle retention, which is a direction for future improvements. As shown in Figure 3 and Table 2 of the supplementary PDF, we supplemented qualitative and quantitative experiments to demonstrate our method's ability to preserve hairstyle details.
>
> **W3:** When the target hairstyle is the one generated by the Hairstyle Editing Stage,I_c is P^s, while retaining the original hairstyle means I_i. This aspect was not clearly explained. The style proxy is intended to guide images with the target hairstyle, as mentioned in the introduction. Therefore, when only changing the hair color while retaining the original hairstyle, P^s in Figure 2 will be replaced by I_i. We will address this in future versions. Thank you for your help!

---

> > ### Comment · Reviewer_X6RQ · 2024-08-13
> >
> > Thank you for the rebuttal and the detailed explanations provided. While I appreciate the clarifications, I still have concerns regarding the overall novelty of the work. Specifically, the points raised about 'Addressing the lack of paired datasets' and 'Preserving hairstyle structure' do not seem to significantly advance the state of the art.
> >
> > Moreover, in your rebuttal, you mentioned that 'color discrepancy outside the inpainting region is a common challenge in diffusion-based inpainting tasks.' However, in Figure 4, the claim that 'Our approach shows better preservation of irrelevant attributes' appears somewhat overstated given the context. This discrepancy raises questions about the extent of improvement your method offers.
> >
> > Given these concerns, I have decided to maintain my original rating. That said, I am open to reconsideration if other reviewers strongly advocate for the acceptance of this paper.

---

> > > ### Author Response · Authors · 2024-08-14
> > >
> > > Thank you very much for taking the time to review my work amidst your busy schedule.
> > >
> > > **Innovativeness and Effectiveness** : Our approach has achieved notable results in the novel task of multi-color hair transfer. Previously, no method applied the warp model to hair color transfer. We believe that if existing methods in the hairstyle domain had utilized the warp module, it would be necessary to compare our approach with them and demonstrate that our method surpasses them in metrics. However, as this is our first introduction of this module, we have validated the effectiveness of our method through the following:
> > >
> > > - Ablation Study (pdf Table 1): We have conducted ablation studies to verify the effectiveness of each step and the integration of the warp module.
> > > - Evaluation Metrics (pdf Table 2): As shown in Table 2 of the supplementary materials, our method, based on a general hair editing self-transfer approach, outperforms previous methods in the task of multi-color hair transfer. While our metrics in the task of swapping monochrome hairstyles are not the best, they are comparable to earlier methods. Considering our focus on the multi-color task, we believe this is justified.
> > >
> > >
> > > **Preservation of Facial Details and Irrelevant Attributes:**  We have not overstated our effectiveness in preserving irrelevant attributes. In fact, in most cases, our method excels in maintaining facial details, accessories, clothing, and background. Although there are some changes in skin tone on the left side of Figure 4, the retention of facial details and the background is excellent. For example, in Figure 5 of the paper, despite changes in skin tone, the clothes of the child behind are preserved much better than in the text2img comparison method. Therefore, from the overall quantitative metrics and the preservation of irrelevant attributes, our method is superior to the comparison methods. We are willing to provide additional image examples to further demonstrate our preservation of details in the background, clothing, and other aspects.
> > >
> > > I greatly value your opinions and hope to improve my research under your guidance.
> > >
> > > Thank you for your assistance.

---

> ### Author Response · Authors · 2024-08-13
>
> Due to the upcoming deadline for the discussion, I would like to confirm if you have any further questions or if there are areas where you need further clarification. If there are no more issues with my submission, I hope to receive your feedback and proceed.
>
> Thank you very much for taking the time to review my work amidst your busy schedule. I greatly value your opinions and hope to improve my research under your guidance.
>
> Thank you for your assistance, and I look forward to your reply.

---

### Official Review · Reviewer_Ubef · 2024-07-11

**Soundness:** 3
**Presentation:** 3
**Contribution:** 3
**Rating:** 6
**Confidence:** 4

**Summary:**

The paper introduces an approach for hair editing using Latent Diffusion Models (LDMs). A warping module ensures precise alignment of the target hair mask and enables hair color structure editing using reference images. The proposed Multi-Hair Basis (MHB) method within LDMs decouples hair color and hairstyle. The authors showcase the performance of their method through GAN-based methods via qualitative and quantitative evaluations,

**Strengths:**

The paper introduces a warping module ensuring precise alignment of the target hair mask. This helps in handling some small mismatches between the images.

The method separates hair color from hairstyle. This allows flexibility and control over the hair transfer task.

The paper also shows results of text-based hairstyle editing, reference image-based hair color editing, and claims to preserve facial attributes.

**Weaknesses:**

1) Most of the examples shown in the results are front-facing subjects in the case of the hairstyle transfer. In real-world applications, there are cases where the reference and source images can have diverse/different poses. The lack of such results raises the question if this is a limitation of the method. Such problems are addressed in the paper HairNet[1].

2) A limitation of some GAN-based approaches compared in the paper is that to achieve high-fidelity reconstruction, the images are overfitted into the networks. As such most of these methods do not support further edits, for example shortening the transferred hair, making it wavy or curly or changing the pose of the subject to view the subject from a different angle. The current method does not seem to solve this problem either. What are the advantages of these GAN frameworks? Some of these methods support further editing as well. (check video associated with HairNet). https://www.youtube.com/watch?v=WBB43cgCFZM&t=153s

3) Some of the GAN based approaches can also control the degree of hairstyle transfer (Barbershop). Does this method control this efficiently?

[1] Zhu, Peihao, et al. "Hairnet: Hairstyle transfer with pose changes." European Conference on Computer Vision. Cham: Springer Nature Switzerland, 2022.

**Questions:**

1) Most examples in the results are front-facing subjects. How does the method perform when the reference and source images have diverse or significantly different poses?

2) GAN-based approaches often overfit images for high-fidelity reconstruction, limiting further edits. How does the current method handle this issue?

3) Can your method support additional edits post-hairstyle transfer, such as shortening hair, making it wavy or curly, or changing the subject's pose to view from different angles?

4) What specific advantages does the current method offer over existing GAN frameworks, especially regarding editability and handling diverse poses?

Also, check the weakness section.

**Limitations:**

The authors have discussed the limitations.

---

> ### Author Rebuttal · Authors · 2024-08-07
>
> Thank you to the reviewers for pointing out the weaknesses.
> 1. We have reviewed the HairNet video and paper. The issue of transferring hair across diverse/different poses is not the problem addressed in our paper. HairNet can effectively control different facial angles, which would be highly beneficial for our method to achieve multi-angle and multi-color hair transfer in the future. We will cite this work in our revised version. However, it is challenging for GAN-based methods to excel in both editability and retention of original features. In Figure 5 of our parper, we compare various space mapping methods based on StyleGAN (including Barbershop based on FS space), and it is evident that they struggle to reconstruct such niche features as multi-color hairstyles. In the global response,  we present some examples of our Warping module handling cases with significantly different poses.
> 2. **Advantages of GAN-based methods**:
>   1. The speed of image generation is very fast. Benefiting from the decoupled facial features in StyleGAN, it is almost possible to change certain feature vectors and quickly generate the corresponding images.
>   2. They retain editability, allowing operations in the latent space, where feature vectors in the StyleGAN latent space can be edited to control different features such as hair length and face orientation.
> 3. Barbershop cannot control multi-color hair transfer. Although the FS space based on optimization can retain facial feature details, it is difficult to decouple the two features of hair color and hair color structure, because these two features are highly coupled (Figure 5 in the parper compares Barbershop).
>
> Below are the responses to the questions:
>
> **A1:** The global response section G1  discussion on the Warping module's ability to transfer hair color under extreme conditions, including significant posture differences, complex textures, and large differences in hair regions.
>
> **A2:** Diffusion models generate images through a multi-step denoising process, from pure noise to clear images. This allows the model to better capture image details and semantic information at different levels during the generation process, rather than generating the entire image at once. This gradual generation approach helps avoid overfitting, as each step improves and adjusts different levels of the image. This editing capability makes diffusion models more flexible and controllable for different editing tasks. Our paper adopts several common Diffusion Inpainting methods to ensure the masked region remains unchanged. For example, we input the Hair Agnostic Mask and Source Image together into the model, allowing it to distinguish between known and unknown areas, thus only filling in the unknown areas without affecting the known ones. During the generation process, noise can be injected only into the masked areas while keeping the unmasked areas unaffected. This means the unmasked areas always retain their original pixel values during the diffusion process, ensuring these areas are not influenced by the generation process. For the hairstyle region needing editing, we remove information from the original image other than the face, use sparse Controlnet's openpose and textual information to control the hairstyle generation StyleProxy, then further obtain the hairstyle region mask of StyleProxy. As mentioned above, we input these together into the model, thereby preserving both the editability of the hairstyle and the high fidelity of the unrelated regions.
>
> **A3:** Our method does not support additional edits post-hairstyle transfer. Using StyleGAN, direct operations in the latent space can achieve quick and direct feature editing. However, the process of generating images via Diffusion is a multi-step gradual denoising process, where the latent space is not directly editable but involves a series of denoising steps to achieve image generation. Therefore, direct editing in the latent space is more challenging. The generation process of diffusion models requires multiple iterative steps, making real-time editing as in StyleGAN's HairNet impractical. Although some degree of editing can be achieved through conditional generation, such editing usually needs pre-set conditions before generation, making it difficult to achieve real-time, dynamic feature adjustments.
>
> **A4:**
> - **Editability**:
> 1. For text-based hairstyle editing, compared to the current HairCLIP and HairCLIPv2 based on the combination of CLIP and StyleGAN, their editing capabilities are limited by the dimensions of the StyleGAN latent space and the GAN training process. They cannot capture niche and complex facial features or information outside the face, such as background and arms. In contrast, diffusion models can combine text conditions multiple times during the generation process, ensuring that each step of image generation aligns with the text description. This method better captures the details and complexity in the text description.
> 2. Our method can integrate multiple conditions for additional control, such as stroke maps to control hair color, controlnet to introduce canny maps to control hair structure, and denspose maps to ensure the hairstyle generation aligns with facial posture. These capabilities significantly enhance the editability of our method.
>
> - **Handling pose diversity**: Compared to methods specifically dealing with multi-pose hair transfer, our paper focuses more on color transfer while maintaining the original hairstyle. The warping module aligning the target hair color and region with different facial poses is a conditional GAN method, as it is currently challenging to decouple hairstyle and hair color directly in the latent space of diffusion models. There are also related articles exploring multi-view generation in diffusion models, such as DiffPortrait3D, which achieves multi-view facial effects.

---

### Official Review · Reviewer_8NSD · 2024-07-12

**Soundness:** 1
**Presentation:** 2
**Contribution:** 1
**Rating:** 4
**Confidence:** 5

**Summary:**

This paper presents a novel approach called HairDiffusion for editing hair in images using latent diffusion models. The main contributions of the work are:

1. Introduction of the Multi-stage Hairstyle Blend (MHB) method for effectively separating control over hair color and hairstyle in the latent space of the diffusion model. MHB divides the diffusion process into two stages, allowing for precise guidance of hair color generation and context-aware generation of the rest of the image.

2. Development of a warping module to align hair color with the target area. This module adapts the HR-VITON architecture, using DensePose and segmentation maps to account for facial poses.

3. Utilization of hair-agnostic masks to transform the hair editing task into an inpainting task. The authors developed two types of masks for different editing stages, which effectively preserve necessary information and remove unnecessary information.

4. Fine-tuning of the CLIP model on a dataset with multi-colored hairstyles to improve editing of complex hair color structures. The authors applied data augmentation to increase pattern diversity.

The authors demonstrate that their method outperforms existing approaches in hairstyle and hair color editing tasks, especially for complex multi-colored styles. The method allows for editing hairstyle and hair color separately or together, using textual descriptions or reference images. Experiments show that HairDiffusion better preserves relevant image attributes (e.g., facial features, background) compared to existing methods.

The integration of these components into a single pipeline enables HairDiffusion to work effectively with both simple and complex multi-colored hairstyles while preserving other image attributes.

**Strengths:**

A major strength of this method is the quality of complex hair color transfer, as well as the preservation of hair texture, which other methods typically don't pay much attention to. The approach demonstrates exceptional ability in handling multi-colored hairstyles. Additionally, the method's capacity to preserve relevant attributes such as facial features and background elements sets it apart from existing techniques. The flexibility to edit hairstyle and color separately or together, using either text or reference images, adds to its versatility and practical applicability.

**Weaknesses:**

The paper exhibits several weaknesses. Firstly, it employs confusing notations, some of which are not properly introduced in the text. Secondly, the ablation study lacks clarity and sufficient detail. There is also a notable absence of comprehensive information regarding the method's limitations. For instance, the paper fails to provide examples demonstrating how the style transfer functions when transferring from long to short hair, how it handles cases with significant pose differences, or how it manages complex textures and hairstyles. Moreover, there is insufficient information about the limitations of the warping module. All presented images showcasing this module display identical hair shapes, suggesting that the warping module's functionality is not truly tested. This makes it impossible to accurately evaluate the quality of its performance.

Furthermore, the paper lacks sufficient information for its reproduction, as it wasn't explained how exactly and on what data the CLIP model was fine-tuned. In general, there is very little information throughout the work about additional datasets, where and how the data was scraped, and about hyperparameters.

The user study is very poorly described; there is no information anywhere about what exactly the questions looked like and in which domains these experiments were conducted. There are also questions about the statistical significance of the results.

The scientific novelty of this work is questionable. As demonstrated in Figure 3, the Control-SD method produces hairstyles that are virtually identical to those generated by HairDiffusion. This similarity suggests that the main contribution of the work is specifically in the transfer of hair color rather than the creation of the hairstyle. The color transfer component primarily consists of two stage: the warping module (which is based on HR-VITON, published in 2022) and the Multi-stage Hairstyle Blend (MHB) method (which utilizes ControlNet and simple blending techniques). Given that these core components are largely derived from existing methods, the overall originality of the approach is limited.

Overall, the paper suffers from a lack of comprehensive evaluation. It presents a limited range of visual comparisons and metrics, which hinders a comprehensive evaluation of the method's performance. Specifically, it lacks crucial metrics such as realism after editing (FID), realism with pose differences (FID), and full-image reconstruction metrics (PSNR, SSIM, LPIPS, FID, RMSE) as used in [1]. Moreover, the authors have not included comparisons with recent relevant works like StyleGANSalon [1] and HairFastGAN [2]. This omission of key comparisons and metrics significantly limits the reader's ability to fully assess the performance of the proposed method in relation to the current state-of-the-art approaches.

[1] Sasikarn Khwanmuang, Pakkapon Phongthawee, Patsorn Sangkloy, Supasorn Suwajanakorn. StyleGAN Salon: Multi-View Latent Optimization for Pose-Invariant Hairstyle Transfer. arXiv preprint arXiv:2304.02744, 2023.

[2] Maxim Nikolaev, Mikhail Kuznetsov, Dmitry Vetrov, Aibek Alanov. HairFastGAN: Realistic and Robust Hair Transfer with a Fast Encoder-Based Approach. arXiv preprint arXiv:2404.01094, 2024.

**Questions:**

1. As a strong point of your method, you indicate excellent quality in preserving facial details, but you compare it with HairCLIP v2, which works in a relatively low-dimensional FS space in StyleGAN. Could you provide more examples of facial detail preservation on very complex images in comparison with methods like StyleGANSalon and HairFastGAN?

2. Could you provide more examples of the limitations of your method? We would like to see how the method works with large pose differences, complex textures, and very different hair shapes in the image domain. With these images, we would like to understand how well the concept preservation works in the CLIP space, how well the warp module works, and how well the entire method functions overall.

3. Could you provide information on how you collected the additional dataset with hair colors?

**Limitations:**

The paper does not adequately address the limitations of the proposed method. The authors dedicate only two brief paragraphs to limitations, and these are relegated to the supplementary materials. This treatment is insufficient for a comprehensive understanding of the method's constraints. A more appropriate approach would be to include a detailed discussion of limitations in the main body of the paper. Furthermore, the authors should present visual examples demonstrating scenarios where each module, as well as the overall method, underperforms. These examples should be accompanied by in-depth analysis to provide insights into the reasons for these limitations and potential avenues for future improvements. Such transparency would significantly enhance the paper's scientific value and reproducibility.

---

> ### Author Rebuttal · Authors · 2024-08-07
>
> Thank you for your valuable feedback.
> 1. We will correct the typos and missing text, and supplement the finetune hyperparameter settings in the future version.
> 2. We have showed the limitations of our method's dependency on masks in the supplementary materials. We have supplemented the limitations and ablation experiments proving the effectiveness of the warping module (Figure 1, Table 2).
> 3. The results of the ablation experiments in our paper clearly demonstrate the considerations at each step of our pipeline, and the reasons have been analyzed.
> 4. User Study strictly follows HairCLIPv2.
> 5. We have supplemented the experiments with HairFastGAN metrics and detailed visual comparisons with HairFastGAN and StyleGANSalon in the global response sections G3, G4 and G5. The following outlines reasons for not making certain comparisons:
>    - Quantitative comparisons with StyleGANSalon: StyleGANSalon is based on EG3D pre-trained on the FFHQ dataset for pose transfer, and EG3D does not provide pre-trained models on CelebA-HQ. Hence, we do not conduct quantitative metric comparisons with StyleGANSalon. Moreover, StyleGANSalon performs comparisons on two subsets of the FFHQ dataset without disclosing the partitioning method.
>    - Quantitative self-transfer comparisons with StyleGANSalon: Unlike HairFastGAN and our method, which can decouple and transfer hair color and hairstyle, StyleGANSalon transfers the entire hairstyle. Thus, we believe that comparisons would be unfair.
>    - Qualitative color transfer comparisons with StyleGANSalon: StyleGANSalon transfers the entire hairstyle and cannot decouple and transfer hair color independently, making hair color transfer visual comparisons infeasible.
>
> Regarding the concern that "the Control-SD method produces hairstyles that are virtually identical to those generated by HairDiffusion": The advantage of our method over diffusion-based text2img lies in multi-color text control and the maintenance of the original hair color when only using text to control the hairstyle.
> Our work is the first to introduce a warping module for color alignment in the field of hairstyle editing, differing from previous StyleGAN-based methods. This approach faithfully aligns with the target hair color without being constrained by the limitations of the StyleGAN latent space. To achieve color alignment through the warping module, we undertook the following steps:
> 1. Addressing the lack of paired datasets: We used semantic segmentation and data augmentation to obtain hair regions deviating from the original face.
> 2. Preserving hairstyle details: We employed bilateral filtering to eliminate low-frequency details, improving the quality of hairstyle generation.
> Overall, our approach can guide future researchers in applying the warping module to hairstyle editing.
> The hair-agnostic masks we proposed is crucial for enabling hairstyle editing tasks with diffusion methods. It maximizes the preservation of hairstyle-independent attributes while retaining the editability of the hairstyle region. Although our pipeline borrows certain components from existing techniques, we focus on integrating and optimizing these techniques to address specific issues in practical applications. For example, achieving a balance between hair color and hairstyle blending while minimizing artifacts, filling in blank regions without hair, improving the quality of generated hairstyles, and achieving more flexible multi-condition editing through the multi-step decoupling of hairstyle and hair color are all considerations in practical applications.
>
> Responses to specific questions are as follows:
>
> **A1:** In the global response sections G3, G4 and G5, we have supplemented experiments with HairFastGAN and StyleGANSalon, including detailed visual comparisons in self-transfer methods with HairFastGAN and StyleGANSalon.
> As shown in Figure 3, our method demonstrates better visual effects in preserving multi-color hairstyles and hairstyle details under facial detail attributes and occlusion conditions. Since whole hair transfer is not the focus of our work and StyleGANSalon is a pose transfer model pre-trained on EG3D with the FFHQ dataset, we do not conduct quantitative metric comparisons with StyleGANSalon.
>
> **A2:**
> - 1. **Limitations of the Warping Module**: We have supplemented discussions on the limitations of the warping module in extreme cases of hair color transfer in Figure 1 of the supplementary PDF, including significant pose differences, complex textures, and large differences in hairstyle regions.
> - 2. **Effectiveness of the Warping Module**: We have added an ablation experiment where 2000 images from the CelebA-HQ validation set are divided into source images and target hair color images for mutual color transfer experiments. The results are shown in Table 1 of the supplementary PDF.
> As for Effectiveness of CLIP Fine-tuning, the results are shown in section G6 of the supplementary PDF.
>
> **A3:**
> Dataset Details: Number : 4,625;
>
> Average size: 224.93 x 264.25;
>
> Categories:
> - Hair colors: purple, red, green, blue, brown, silver, blonde, pink, burgundy;
> - Color structures: ombre, dip dyed, streaked, half and half, split color.
>
> Data acquisition: We crawled tens of thousands of multi-color hair images using keywords, cleaned the data, and obtained approximately six thousand multi-color hair images. These images were input into GPT with predefined hair color and structure categories to generate text annotations like "a photo of {colors}, {color structure} hairstyle." We then had 10 professional annotators rate the annotations in multiple rounds, discarding low-rated images, resulting in 4,625 multi-color text-image pairs.  If the paper is accepted, we will make this dataset publicly available.

---

> > ### Comment · Reviewer_8NSD · 2024-08-10
> >
> > Thank you for your responses. However, there are still some concerns regarding the limitations of your method that were not fully addressed in the rebuttal.
> >
> > Regarding your answer A1, you demonstrate an improved reconstruction by referencing Table 2 and Figure 3. While this shows that your method outperforms in reconstruction tasks, it was already evident that Stable Diffusion's reconstruction is superior to StyleGAN's. However, Table 2 indicates that your method underperforms significantly compared to StyleGAN-based methods in single color transfer. It would be beneficial to explain why this occurs and provide examples to illustrate this point. Additionally, the impact of such transfers on facial details remains unclear and should be elaborated upon.
> >
> > Your answer A2 did not directly address my original question. I inquired about how your method performs when transferring hairstyles from the image domain and its general limitations. Instead, you focused on the limitations of your method when recoloring hair.
> >
> > In summary, the current presentation of the work lacks transparency, and there are several issues that need to be addressed. If the final paper is permitted to incorporate these important responses from the rebuttal, I believe it would be suitable for publication. Additionally, given the recent publication of the Stable-Hair paper [1], it would be valuable to include a comparison with this work in the future version of your paper.
> >
> > [1] Yuxuan Zhang, Qing Zhang, Yiren Song, Jiaming Liu, Stable-Hair: Real-World Hair Transfer via Diffusion Model, arXiv:2407.14078

---

> > > ### Author Response · Authors · 2024-08-12
> > > **The discussion the challenges and limitations of the current method in hairstyle transfer, color alignment, and retaining facial details, while suggesting future improvements and comparisons with other approaches.**
> > >
> > > Thank you for your suggestions, which have been very helpful in improving our paper.
> > > Discussion on the Performance Compared to StyleGAN-Based Methods
> > > 1. Reasons for Subpar Performance Compared to StyleGAN-Based Methods:
> > >   - In cases where hairstyles differ significantly, our color alignment method, which is designed to generate colors more consistent with adjacent hues, may produce colors that deviate from the original hairstyle color. This discrepancy can lead to inconsistencies in the generated results.
> > >   - The sparse matrix control provided by ControlNet's Canny map does not offer pixel-to-pixel control over hairstyle features. Alternatively, we could attempt to use a step-by-step approach that mixes multiple ControlNet models, allowing the model to focus on additional information.
> > >   - When editing hairstyles based on reference images while retaining Diffusion text editing, setting the text to "hair" by default may introduce additional information that degrades the quality of the generated results. We could try separating the control of the text and ControlNet to balance the control over hairstyle structure and color in the future.
> > >   We will include examples of these issues in future versions.
> > > Impact on Facial Details
> > > 2. Transfers and Facial Details:
> > >   - As illustrated in Figure 3, StyleGAN-based methods show suboptimal retention of accessories (G, M, O) and unique facial features (I) when the training dataset contains few or no samples of these details. The preservation of multi-color hair (N, Q) is compromised because HairFastGAN's approach of decoupling hair color and hairstyle in the feature space makes it challenging to maintain the color structure. StyleGAN Salon employs post-processing enhancements that improve results for multi-color hair to some extent, but retaining hairstyle details remains challenging. Hairstyle preservation (A, B, C, E, T, S, R, N) suffers from the lack of masks to retain irrelevant information. Additionally, maintaining the background (P) and hand preservation (F, K, L, W) in cases where the hand obscures the hairstyle is also difficult without optimization. StyleGAN-based methods struggle to preserve detail for accessories or hands, particularly when the training dataset contains few examples of such features.
> > >   - Conversely, the ControlNet Canny map, which provides line-based control, shows better performance in preserving hairstyle flow and individual hair strands.
> > > Discussion on Hairstyle Transfer Limitations
> > > 3. Limitations of Hairstyle Transfer:
> > >   - Hairstyle transfer is not the primary focus of our work, but it is important to address the method's shortcomings. We use a straightforward approach of converting hairstyles into text vectors, which often fails to capture the complete details and local features of hairstyles, leading to noticeable discrepancies between the generated and original hairstyles. Recent studies suggest using encoders trained on face datasets to better capture facial details, which is a promising direction for future work. We will include additional figures and textual explanations to discuss these limitations in future versions.
> > >   - We have also noted the paper on Stable-Hair, which differs from our approach by not decoupling hairstyle features, thereby preserving both hair color and overall hairstyle during the transfer, and not focusing on text-based editing of hairstyles. This paper has not yet released code or demos; we will conduct a comparison once these resources are available.

---

> > > ### Author Response · Authors · 2024-08-13
> > >
> > > Due to the upcoming deadline for the discussion, I would like to confirm if you have any further questions or if there are areas where you need further clarification. If there are no more issues with my submission, I hope to receive your feedback and proceed.
> > >
> > > Thank you very much for taking the time to review my work amidst your busy schedule. I greatly value your opinions and hope to improve my research under your guidance.
> > >
> > > Thank you for your assistance, and I look forward to your reply.

---

### Author Rebuttal · Authors · 2024-08-07

**G1: More Examples of the Limitations:** In Figure 1 of the PDF, we have added discussions on the limitations of the warping module in extreme cases of hair color transfer, including significant pose differences, complex textures, and large discrepancies in hairstyle regions. To visually illustrate the limitations of the warping module method, the warped results have not undergone the post-processing mentioned in the paper.
The first and second rows demonstrate cases with significant differences in hair length. While the hairline region aligns well, the hair ends do not; The second row on the left, where the hair lengths are similar, performs well. This is due to the consistent inclusion of hair end information when generating the paired hair dataset used to train the warping module. The third, fourth, and fifth rows show cases with significant pose differences. The warped result on the right side of the second row shows that the hair orientation aligns well with the facial orientation of the Source Image, indicating some robustness in the model across different poses. However, the right side of the second row and the third row still show misalignment in hair parting, leading to color inconsistencies in the Output. The left side of the fourth row shows the model discarding the hair ends while supplementing the bangs area of the Source Image, though there is still some deviation in the hair color's centerline. The last row depicts  complex textures scenarios where the Output hair color does not match the Target Image, due to the compression of images required for diffusion input and the bilateral filtering operation removing high-frequency color details.
For cases with missing target region hair color, post-processing with PatchMatch can fill in the blank areas, as shown in Figure 4. The effectiveness is demonstrated in the ablation experiments in Table 1.


**G2: More Examples of Reconstruction:** In Figure 3, we have supplemented self-transfer experiments with state-of-the-art methods HairFastGAN and StyleGAN Salon. The comparison illustrates the retention of accessories (G, M, O); multi-color hair preservation (N, Q); hand preservation (F, K, L, W); unique facial features retention (I); hairstyle preservation (A, B, C, E, T, S, R, N); background retention (P). Our method also has limitations, such as color discrepancies in attributes other than the hairstyle in some cases, like hair color in the second column and skin color in the seventh column. This is due to merging the noisy latent, masked image latent, stroke map latent, and mask at the initial convolution layer of the UNet architecture, where they are collectively influenced by the text embedding. Consequently, subsequent layers in the UNet model struggle to obtain pure masked image features due to the influence of the text and stroke map.


**G3: Hair Color Transfer Metrics:** In Table 2 left, we referenced HairFastGAN for **single color transfer** metrics testing, showing that our method performs comparably to GAN-based methods on the CelebA-HQ dataset, which predominantly features single-colored hair. However, the quality of the multi-colored hairstyles we crawled supports the fine-tuning of the CLIP model, but the majority do not reach the size of CelebA-HQ (1024x1024), and the data volume is insufficient for quantitative experiments. Thus, we only provide visual comparisons in the paper.


**G4: Hair Reconstruction Metrics:** In Table 2 right, we have referenced HairFastGAN to include more comprehensive metrics for Reconstruction quality, demonstrating the superiority of our method in overall image quality.


**G5: Comparison to HairFastGAN:** In Figure 2, our method outperforms HairFastGAN in both facial preservation and hair color transfer.


**G6:Effectiveness of CLIP Fine-tuning:** In Figure 5, We conducted a simple text-to-image alignment experiment. It demonstrates that our fine-tuned CLIP model performs better in hair color and multi-color structure compared to the original model. Some colors show overfitting, which we plan to address by balancing color proportions in the fine-tuning dataset in future work.

---

### Decision · Program_Chairs · 2024-09-25

**Decision:**

Accept (poster)

**Comment:**

The submission received diverse scores, two positive and two negative. The authors provided detailed rebuttal and discussions with the reviewers. After carefully reading all comments, the AC found only one major concern is remaining: the lacking of novelty. After taking a look at the paper and the AC think the proposed method, though not that novel, is indeed a new thing and it really contributed new performance on hair editing. Given one negative reviewer did not give final responses on the authors' rebuttal and another negative reviewer said he will not argue for acceptance. The AC finally recommended an acceptance.